# NAD+ dependent UPRmt activation underlies intestinal aging caused by mitochondrial DNA mutations

Liang Yang[1,2,3,8], Zifeng Ruan[1,3,4,8], Xiaobing Lin[1,3,8], Hao Wang[1,3], Yanmin Xin[1,3], Haite Tang[1,3], Zhijuan Hu[1,3,4], Yunhao Zhou[1,3,4], Yi Wu [1,2,3], Junwei Wang[1,3], Dajiang Qin[2,5], Gang Lu[6], Kerry M. Loomes[7], Wai-Yee Chan [6] & Xingguo Liu [1,2,3] ✉

Aging in mammals is accompanied by an imbalance of intestinal homeostasis and accumulation of mitochondrial DNA (mtDNA) mutations. However, little is known about how accumulated mtDNA mutations modulate intestinal homeostasis. We observe the accumulation of mtDNA mutations in the small intestine of aged male mice, suggesting an association with physiological intestinal aging. Using polymerase gamma (POLG) mutator mice and wild-type mice, we generate male mice with progressive mtDNA mutation burdens. Investigation utilizing organoid technology and in vivo intestinal stem cell labeling reveals decreased colony formation efficiency of intestinal crypts and LGR5-expressing intestinal stem cells in response to a threshold mtDNA mutation burden. Mechanistically, increased mtDNA mutation burden exacerbates the aging phenotype of the small intestine through ATF5 dependent mitochondrial unfolded protein response (UPRmt) activation. This aging phenotype is reversed by supplementation with the NAD+ precursor, NMN. Thus, we uncover a NAD+ dependent UPRmt triggered by mtDNA mutations that regulates the intestinal aging.

The adult intestinal epithelium is composed of differentiated cells forming villi and proliferating cells in crypts. Leucine-rich repeat-containing G protein-coupled receptor 5 (LGR5)-expressing intestinal stem cells (ISCs), residing at the base of the crypt, can differentiate to different epithelial cells (enterocytes, Paneth cells, goblet cells, microfold cells, tuft cells, etc.) and regenerate the epithelium[1,2]. These differentiated epithelial cells form orderly intestinal structures to maintain intestinal homeostasis and fundamental functions such as absorption, secretion, barrier and antimicrobial function.

ISCs are supported by a niche of accessory cell types to generate mature epithelial cell types, which fuel intestinal renewal, regeneration and development. These niche cells of ISCs include Gli1+, Foxl1+

[1]CAS Key Laboratory of Regenerative Biology, Joint School of Life Sciences, Guangzhou Institutes of Biomedicine and Health, Chinese Academy of Sciences, Guangzhou, China. [2]Centre for Regenerative Medicine and Health, Hong Kong Institute of Science & Innovation, Chinese Academy of Sciences, Hong Kong SAR, China. [3]Guangdong Provincial Key Laboratory of Stem Cell and Regenerative Medicine, China-New Zealand Joint Laboratory on Biomedicine and Health, CUHK-GIBH Joint Research Laboratory on Stem Cells and Regenerative Medicine, Institute for Stem Cell and Regeneration, Guangzhou Institutes of Biomedicine and Health, Chinese Academy of Sciences, Guangzhou, China. [4]University of Chinese Academy of Sciences, Beijing, China. [5]Key Laboratory of Biological Targeting Diagnosis, Therapy and Rehabilitation of Guangdong Higher Education Institutes, The Fifth Affiliated Hospital of Guangzhou Medical University, Guangzhou, Guangdong, China. [6]CUHK-GIBH Joint Research Laboratory on Stem Cells and Regenerative Medicine, CUHK-Jinan University Key Laboratory for Regenerative Medicine, Ministry of Education, School of Biomedical Sciences, Faculty of Medicine, The Chinese University of Hong Kong, Hong Kong SAR, China. [7]School of Biological Sciences and Institute for Innovation in Biotechnology, University of Auckland, Auckland 1010, New Zealand. [8]These authors contributed equally: Liang Yang, Zifeng Ruan, Xiaobing Lin. ✉e-mail: liu_xingguo@gibh.ac.cn

stromal cells and Paneth cells, and secrete Wnt molecules as self-renewal factors to support intestinal crypts[3–5]. Signaling pathways underlying the development process of intestinal epithelial cells are well studied. Wnt and Notch signals are essential for maintaining stem cell activity and balanced differentiation between secretory and absorptive cell lineages[6,7]. The phosphatidylinositol 3-kinase (PI3K)/protein kinase B (Akt) pathway is also involved in the intestinal differentiation and proliferation[8], while the transforming growth factor beta (TGF-β) pathway negatively regulates growth of normal epithelial cells[9–12]. Notum and Wnt signals play vital roles in the aging process of intestine as marked by reduction of ISC number and alteration of ISC function[13,14]. However, detailed understanding of the signal pathways regulating intestinal aging is lacking.

Mitochondrial DNA (mtDNA) mutations are widely regarded as an important cause of aging and age-associated diseases[15,16] and reportedly induce the aging of multiple organs in mouse such as ovary, heart and liver[17–19]. In particular, the aging phenotypes of tissues with strong energy requirements such as heart and liver tissues are accelerated by excessive mtDNA mutations, which facilitate mitochondrial dysfunction by compromising oxidative phosphorylation[20]. The accumulation of mtDNA mutations in aged human clinical intestinal samples[21–23] indicates that mtDNA stability may also be important for intestinal aging. However, the sub-cellular mechanisms involved are still unknown while the perturbed epithelial cell crosstalk caused by increased mtDNA mutation content remains to be investigated.

mtDNA-mutator (*PolgA^Mut/Mut*) mice exhibit multiple premature aging phenotypes by nine months of age[17,24], and decreased lifespan of 13–15 months[25,26]. As a result, *PolgA^Mut/Mut* mice are widely utilized to investigate the roles of mtDNA mutations in the aging process[27–31]. Using this mouse model, we demonstrate an association between mtDNA mutation content and the intestinal aging phenotypes. We found that increased mtDNA mutation burden triggers an ATF5-dependent mitochondrial unfolded protein response (UPR^mt) by NAD+ depletion. These findings reveal a regulatory mechanism whereby the UPR^mt mediates the intestinal aging phenotype caused by increased mtDNA mutation burden.

## Results

### High mtDNA mutation burden induces an aging-like phenotype in the small intestine

Similar to aged human clinical intestinal samples[21–23], the small intestine of aged mouse intestine, characterized by decreased intestinal crypt number and increased villus length, higher expression of CDKN1A/p21 (a well-known senescence marker) and shorter telomere length (Supplementary Fig. 1a–c), accumulates more mtDNA mutations, primarily low-frequency (less than 0.05) point mutations (Supplementary Fig. 1d, e). This observation demonstrates a link between mtDNA mutation content and physiological intestinal aging.

To investigate the effect of mtDNA mutation content on small intestinal aging, we generated mice with four ascribed categories of mtDNA mutational burden: negligible (WT/WT), low (WT/WT*), moderate (WT/Mut**) and high (Mut/Mut***) as described[18]. We then detected intestinal changes at 3, 8 and 12 months in intestinal sections and using an organoid system (Fig. 1a). Because intestinal crypt number decreases and villus length increases during intestinal aging[14,32–34], we first performed hematoxylin-eosin (H&E) staining to examine intestinal architecture. As compared to WT/WT, WT/WT* mice with theoretically maternally transmitted mtDNA mutations showed no significant difference in intestinal architecture during aging, indicating little effect on the intestinal aging. As compared to WT/WT and WT/WT*, Mut/Mut*** mice exhibited clear aging-like changes and accumulation of apoptotic cells in the small intestine at 8 months, which was more pronounced at 12 months (Fig. 1b, c). The higher expression of CDKN1A/ p21 and shorter telomere length in Mut/Mut*** at 8 months also supported exacerbation of small intestine aging with increased

mtDNA mutation burden (Supplementary Fig. 1f, g). We also used an in vitro intestinal organoid system to reflect ISC function[1,35] and found that organoids derived from Mut/Mut*** mice were fewer and smaller in size versus WT/WT* and WT/Mut** mice (Fig. 1d). The decreased colony formation efficiency in Mut/Mut*** mice indicates mtDNA mutations impair ISC function.

To study the link between the observed aging phenotypes and mtDNA mutation content in detail, we profiled mtDNA mutation types and frequencies in the small intestines of our mouse models. As compared to WT/WT mice, at 8 months Mut/Mut*** mice accumulate significantly more point mutations, primarily as low-frequency (less than 0.05) point mutations; these were also observed in the small intestine of aged wild-type mice (Supplementary Fig. 2a–d). Notably, the intestine of WT/WT* and WT/Mut** mice with little apparent aging phenotype accumulate low quantities of mtDNA mutations (less than 900) at 8 months whereas the intestine of Mut/Mut*** mice accumulates high quantities of mtDNA mutations (average mutation number above 4,000) at 8 months. Interestingly, the brain and liver of WT/Mut** mice did accumulate much more expected mtDNA point mutations but small intestine didn't, implying a possible mechanism for clearing mutated mtDNAs in the small intestine (Supplementary Fig. 2b–d and Supplementary Fig. 3a, b). Overall, the above results indicate that increased mtDNA mutation burden induces an aging-like phenotype in the small intestine.

### mtDNA mutations result in NAD+ depletion during intestinal aging

To further explore the mechanism by which increased mtDNA mutation burden regulates intestinal aging, we performed RNA sequencing (RNA-seq) and analyzed 780 upregulated differential genes in the intestines of Mut/Mut*** mice as compared to WT/WT* at 8 months using Gene Ontology (GO) analysis (Supplementary Fig. 4a, b). Most enriched GO terms are involved in mitochondrial respiratory function and metabolic processes, demonstrating an alteration of mitochondrial function caused by mtDNA mutations. As expected, the intestinal crypt cells in Mut/Mut*** mice exhibit a lower mitochondrial inner membrane potential and higher ROS concentrations as compared to WT/WT* mice, indicating that high mtDNA mutation content results in mitochondrial dysfunction during intestinal aging (Supplementary Fig. 4c, d). Notably, the enriched NADH dehydrogenase complex assembly pathway implies an impairment of NADH/NAD+ redox in Mut/Mut*** intestines (Fig. 2a). This perturbed redox potential was confirmed by transfection of intestinal crypt cells with SoNar, a NADH/NAD+ sensor[36,37], which showed a higher NADH/NAD+ ratio in Mut/Mut*** mice (Fig. 2b, c). We also transfected intestinal crypt cells with FiNad (a NAD+ sensor[38]), and found less NAD+ content in the Mut/Mut*** cells (Fig. 2d, e), leading us to investigate the role of NAD+ depletion in the intestinal aging.

### NAD+ repletion alleviates small intestinal aging caused by increased mtDNA mutation content, which is independent of mitophagy

In response to dietary supplementation with the NAD+ precursor, nicotinamide mononucleotide (NMN)[39,40], we found increased crypt number, decreased villus length and presence of apoptotic cells in the intestine of Mut/Mut*** mice (Fig. 3a, b). We further validated the upregulation of NAD+ concentration in the intestinal crypts of Mut/Mut*** mice treated with NMN (Supplementary Fig. 5), thus indicating that NAD+ repletion in vivo alleviates the small intestine aging phenotypes caused by mtDNA mutation burden. In parallel, we found NAD+ repletion in vitro could rescue decreased colony formation efficiency in Mut/Mut*** intestinal organoids (Fig. 3c). Together these results indicate that NAD+ depletion is a key mediator by which increased mtDNA mutation burden induces the aging phenotype of small intestine.

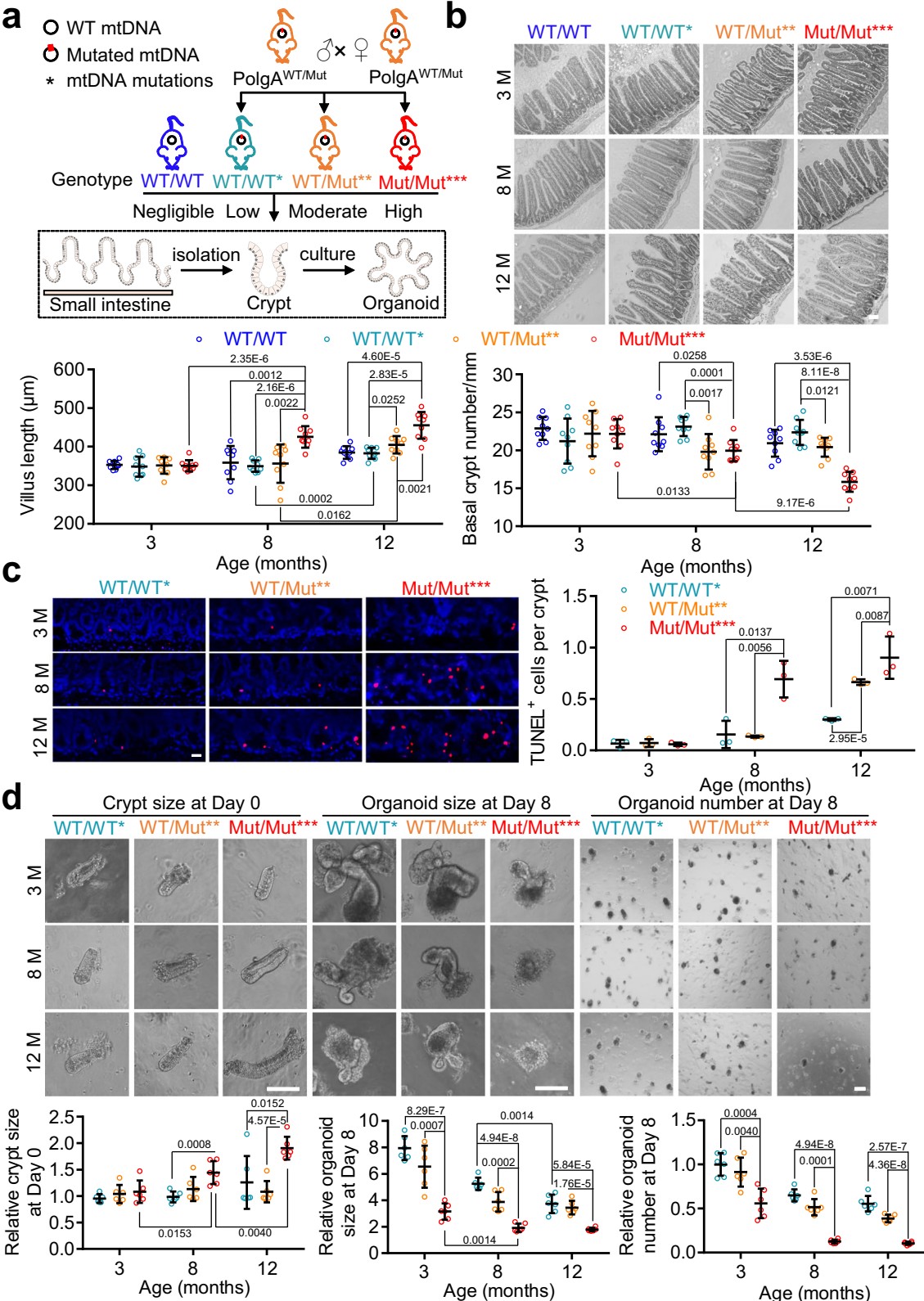

As NAD+ depletion reportedly compromises mitophagy[41,42], it raises the possibility that NAD+ repletion could activate mitophagy to remove mutated mtDNAs, thereby alleviating the small intestine aging phenotypes. By detecting mitophagy using anti-LC3B and anti-TOM20 IF in the intestine crypts of WT/WT*, WT/Mut** and Mut/Mut*** mice at 3, 8 and 12 months, we indeed observed compromised mitophagy in Mut/Mut*** mice at 8 months (Supplementary Fig. 6a). Nevertheless, NAD+ repletion in vivo did not rescue mitophagy deficiency or mtDNA mutation burden in the intestinal crypts of Mut/Mut*** mice at 8 months (Supplementary Fig. 6b, c), indicating that intestinal aging induced by NAD+ depletion is independent of mitophagy.

**Fig. 1 | Increased mtDNA mutation burden induces an aging-like phenotype in the small intestine. a** Diagram depicting the experimental methods for detecting the changes of small intestine and organoids derived from isolated crypts in the mice with four categories of mtDNA mutation burden at 3, 8, and 12 month. The figures of small intestine, intestinal crypt, and organoids were generated with the help of Servier Medical Art, provided by Servier, licensed under a Creative Commons Attribution 4.0 unported license (https://creativecommons.org/licenses/by/4.0/deed.en). *PolgA^{WT/Mut}* (WT/Mut**) mice were crossed to generate three types of offspring with a theoretically maternally transmitted mtDNA mutations, i.e. *PolgA^{WT/WT}* (WT/WT*), *PolgA^{WT/Mut}* (WT/Mut**) and *PolgA^{Mut/Mut}* (Mut/Mut***). *PolgA^{WT/WT}* mice with negligible level of mtDNA mutations (WT/WT) were used as an additional control. **b** Representative images of small intestine stained with H&E in WT/WT, WT/WT*, WT/Mut**, and Mut/Mut*** mice at 3, 8, and 12 months of age (Scale bar, 100 μm). The villus length and number of crypts per millimeter of small intestine are quantified below (Data are presented as the mean ± S.D of 9 intestinal segments from 3 mice per group; two-way ANOVA test). **c** Representative images of intestinal crypts stained with TUNEL in WT/WT*, WT/Mut**, and Mut/Mut*** mice at 3, 8, and 12 months of age (Scale bar, 20 μm). Since no difference was observed during intestinal aging between WT/WT and WT/WT*, WT/WT* mice was used a control. The number of TUNEL-positive cells per crypt is quantified on right (Data are presented as the mean ± S.D and *n* = 3 mice per group; two-way ANOVA test). **d** Representative crypt images at Day 0 and organoids at Day 8 in the intestine of WT/WT*, WT/Mut**, and Mut/Mut*** mice at 3, 8, and 12 months of age (Scale bar, 100 μm). Crypt size, organoid size and relative organoid number per well were quantified. Average organoid number per well was normalized to WT/WT* at 3 months. Data are presented as the mean ± S.D of 6 wells from 3 mice per group; two-way ANOVA test. Source data are provided with this paper.

## Increased mtDNA mutation burden impairs Wnt/β-catenin signaling and induces a decline in LGR5-positive intestinal cells through NAD+ depletion

Due to the important roles of signaling pathways including Wnt, Notch, Notum, PI3K/Akt, and TGF-β pathways in regulating the development process of intestinal epithelial cells, we assayed these signaling pathways and their correlation with NAD+ depletion during intestinal aging. Using RNA-seq analysis, we found that only differential genes associated with Wnt/β-catenin signaling were enriched in Mut/Mut*** versus WT/WT* intestine at 8 months, which are downregulated by mtDNA mutations (Fig. 4a). As Wnt/β-catenin signaling is required for LGR5-expressing ISCs to maintain intestinal regeneration, as marked by the presence of Cyclin D1 (CD1)-expressing cells (Fig. 4b), we further performed anti-β-catenin and anti-CD1 immunofluorescence (IF) in the intestinal crypts of WT/WT*, WT/Mut** and Mut/Mut*** mice at 3, 8 and 12 months. We found that β-catenin fluorescence intensity (FI) is decreased at 8 months with fewer CD1-expressing cells in Mut/Mut*** mice versus either WT/WT* or WT/Mut** mice (Fig. 4c, d). These results suggest that increased mtDNA mutation burden impairs the Wnt/β-catenin pathway. Importantly, NAD+ repletion rescues this pathway in the intestine of Mut/Mut*** mice (Fig. 4e), indicating NAD+ depletion underlies the lack of ISC regeneration through the Wnt/β-catenin pathway.

We next investigated the effect of mtDNA mutation burden on the number of LGR5-expressing ISCs. Crossing LGR5-GFP+/- (LGR5-eGFP-IRES-CreERT2 reporter)[1] and WT/Mut** (*PolgA^{WT/Mut}*) mice, we generated four genotype LGR5-GFP positive mice with increasing levels of mtDNA mutation burden as depicted in Fig. 5a. We noted that LGR5-positive cells decline sharply at 8 months in Mut/Mut*** mice (Fig. 5b), indicating an adverse effect of mtDNA mutation burden. NAD+ repletion increases the number of LGR5-positive intestinal cells in Mut/Mut*** mice (Fig. 5c).

For the requirement of Wnt molecules, such as WNT2[43], WNT4[4] and WNT5A[44], secreted from niche cells including Gli1+, Foxl1+ stromal, Paneth and myofibroblast cells as self-renewal factors to support ISCs[3–5,43,44], we assayed the expression of these Wnt molecules in the intestinal crypts of WT/WT*, WT/Mut** and Mut/Mut*** mice at 8 months. We found all the tested Wnt proteins are decreased in Mut/Mut*** versus WT/WT* mice (Fig. 5d), showing that increased mtDNA mutation burden inhibits secretion of Wnt molecules. NAD+ repletion partially rescues the reduction of WNT2 and WNT4 protein expression in Mut/Mut*** mice (Fig. 5d and Supplementary Fig. 7a, b), implying that mtDNA mutation burden can inhibit the secretion of these two Wnt molecules through NAD+ depletion. Furthermore, we observed downregulation of Foxl1 but not Gli1 in Mut/Mut*** intestine, implying a possible reduction of Foxl1+ cells caused by mtDNA mutation burden (Supplementary Fig. 7c).

As activation of Notch signaling reportedly induces a subset of Paneth cells to gain stem cell features with subsequently proliferation and differentiation into villus epithelial cells[45], we performed anti-Notch1 immunofluorescence. Notch signal activation was observed in the intestinal crypt base of Mut/Mut*** mice at 8 months (Supplementary Fig. 7d), providing a possible explanation for the increase of villus length in the aged intestine lacking the Wnt/β-catenin pathway. NAD+ repletion rescues the Foxl1 downregulation and Notch1 upregulation in Mut/Mut*** mice (Supplementary Fig. 7e), suggesting that mtDNA mutation burden can regulate the function or number of niche cells through NAD+ depletion. Taken together, NAD+ depletion caused by increased mtDNA mutation burden induces the decline of LGR5-positive intestinal cells via impairment of the Wnt/β-catenin pathway.

## NAD+ depletion activates ISR to regulate mtDNA mutation-induced aging phenotypes through impairment of the Wnt/β-catenin pathway

How does NAD+ depletion regulate the Wnt/β-catenin pathway during the intestinal aging caused by mtDNA mutation burden? We performed Kyoto Encyclopedia of Genes and Genomes (KEGG) pathway analysis of the upregulated differential genes in the intestines of Mut/Mut*** versus WT/WT* mice at 8 months. Here, increased mtDNA mutation burden triggers a retrograde mitochondria-to-nucleus communication by upregulating ISR including oxidative phosphorylation, glutathione metabolism, nicotinate and nicotinamide metabolism, folate biosynthesis, the PPAR signaling pathway and carbon metabolism (Fig. 6a and Supplementary Fig. 8a).

Upon mitochondrial ISR, eukaryotic initiation factor 2 α subunit (eIF2α) is phosphorylated[46], which subsequently results in global attenuation of cytosolic translation coincident with preferential translation of mitochondrial stress-responsive transcription factors such as activating transcription factor 4 (ATF4)[47–50], ATF5[51,52] and the C/EBP Homologous Protein (CHOP)[53,54] to restore mitochondrial function. Using western blotting, we observed upregulation of eIF2α phosphorylation (eIF2α-P) and ATF5 but not ATF4 and CHOP in the intestine of Mut/Mut*** mice, which was decreased by in vivo NAD+ repletion (Fig. 6b, c and Supplementary Fig. 8b, c). Consistent with direct control of ATF5 translation by phosphorylation of eIF2α in response to stress[55], eIF2α phosphorylation inhibitor, ISRIB, suppresses ATF5 upregulation in Mut/Mut*** intestine (Fig. 6d). These results showed that NAD+ depletion triggers ATF5-dependent ISR activation.

ISR inhibition increases β-catenin protein expression in the intestinal crypts of Mut/Mut*** mice (Fig. 6e), implying it regulates intestinal aging by restoring Wnt/β-catenin signaling. To test the roles of ISR activation in intestinal aging, we used ISRIB and si*Atf5* to inhibit ISR in Mut/Mut*** intestinal crypts (Supplementary Fig. 8d). We found that both ISRIB and si*Atf5* could obviously rescue decreased organoid formation efficiency caused by increased mtDNA mutation burden (Fig. 6f, g).

We next asked how NAD+ depletion regulates ISR? Among NAD+ consuming enzymes, SIRT7 downregulation caused by NAD+ depletion during aging reportedly regulates the UPR^mt by suppressing mitochondrial ribosomal protein expression[56,57]. Despite no reduction of SIRT7 protein in Mut/Mut*** versus WT/WT* mice (Supplementary

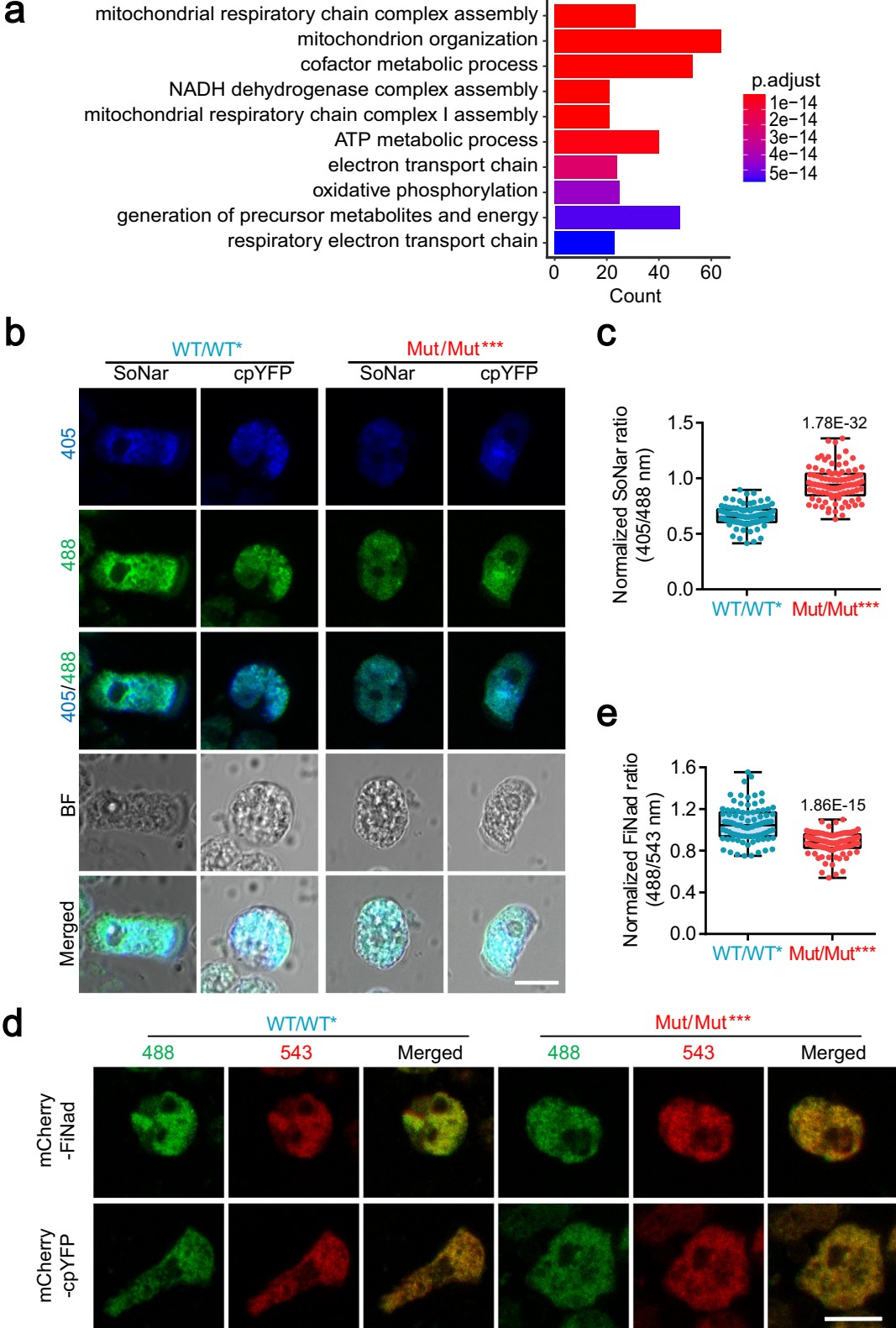

Fig. 8e), SIRT7 overexpression inhibits eIF2α phosphorylation of Mut/Mut*** intestinal crypt cells (Supplementary Fig. 8f). Consistent with previous reports[58,59], these observations support that NAD+ depletion caused by increased mtDNA mutation burden decreases SIRT7 activity to regulate ISR. Taken together, these results indicate that mtDNA mutation burden causes intestinal aging through NAD+ depletion, which activates ATF5-dependent ISR to inhibit Wnt/β-catenin signaling.

## The ISR caused by mtDNA mutation burden regulates UPRmt activation

As the ISR regulation factor, ATF5, activate UPRmt by regulating mitochondrial import efficiency[60], we assayed the marker protein expression of UPRmt including LONP1, caseinolytic peptidase P (ClpP) and heat shock protein 60 (HSP60) in Mut/Mut*** intestine, which exhibited ATF5-dependent ISR activation. Interestingly, only LONP1, a key

**Fig. 2 | Increased mtDNA mutation burden results in NAD+ depletion during the intestinal aging. a** Gene Ontology (GO) enrichment of differentially upregulated genes in Mut/Mut*** mice at 8 months of age versus WT/WT* mice was performed using clusterProfiler package, and significance was determined by Fisher exact test of adjusted *p*-value by Benjamini–Hochberg (BH) method (*n* = 3 mice per group). **b** Detection of NADH/NAD+ ratio in the intestinal crypts of Mut/Mut*** mice and WT/WT* mice at 8 months of age using SoNar and cpYFP imaging (Scale bar, 10 μm). **c** Quantification of NADH/NAD+ ratios as in (**b**) (90 intestinal crypts from 3 mice per group; unpaired comparison using nonparametric Mann–Whitney test).

Central line: median, box: interquartile ranges (IQR), whisker: ranges except extreme outliers (>1.5*IQR), circles: individual data points; **d** Detection of NAD+ content in the intestinal crypts of Mut/Mut*** mice and WT/WT* mice using mCherry-FiNad and mCherry-cpYFP imaging (Scale bar, 10 μm). **e** Quantification of mCherry-FiNad fluorescence corrected by mCherry-cpYFP as in (**d**) (90 intestinal crypts from 3 mice per group; unpaired comparison using nonparametric Mann–Whitney test). Central line: median, box: IQR, whisker: ranges except extreme outliers (>1.5*IQR), circles: individual data points. Source data are provided with this paper.

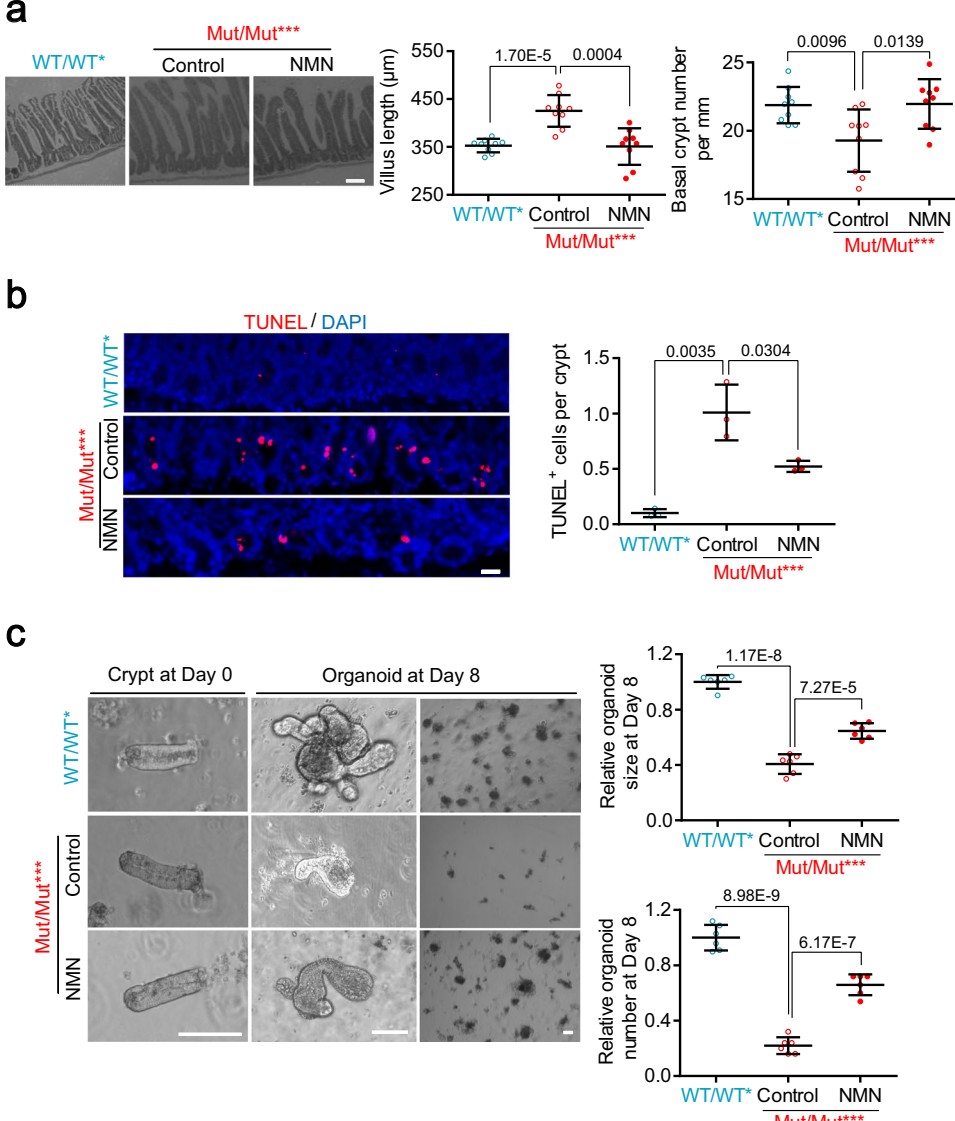

**Fig. 3 | NAD+ repletion alleviates small intestine aging phenotypes caused by mtDNA mutation burden. a** Representative images of small intestine stained with H&E in Mut/Mut*** mice at 8 months of age with NMN or control (water) (Scale bar, 100 μm). Villus length and number of crypts per millimeter are quantified on right; WT/WT* is used as an additional control (Data are presented as the mean ± S.D of 9 intestinal segments from 3 mice per group; unpaired two-tailed Student's *t*-test). **b** Representative images of intestinal crypts stained with TUNEL in Mut/Mut*** mice at 8 months of age with NMN or control (Scale bar, 20 μm). The number of TUNEL-positive cells per crypt is quantified on right (Data are presented as the mean ± S.D and *n* = 3 mice per group; unpaired two-tailed Student's *t*-test. **c** Representative images of crypts at Day 0 and organoids at Day 8 from the intestine of Mut/Mut*** mice at 8 months of age with 200 μM NMN or water control (Scale bar, 100 μm). Organoid size and relative organoid number per well are quantified at bottom. Average organoid number per well is normalized to the control. Data are presented as the mean ± S.D of 6 wells from 3 mice per group; paired two-tailed Student's *t*-test. Source data are provided with this paper.

mitochondrial protease for clearing unfolded proteins upon mitochondrial stress response[61], is obviously increased while its mitochondrial content is not altered in Mut/Mut*** intestinal crypts (Fig. 7a and Supplementary Fig. 9a, b). Both ISRIB and si*Atf5* inhibited the upregulation of LONP1 in Mut/Mut*** intestinal crypt cells (Fig. 7b, c), demonstrating that mtDNA mutation burden regulates LONP1 expression by ATF5-dependent ISR. Overall, intestinal aging induced by increased mtDNA mutation burden is mediated through the

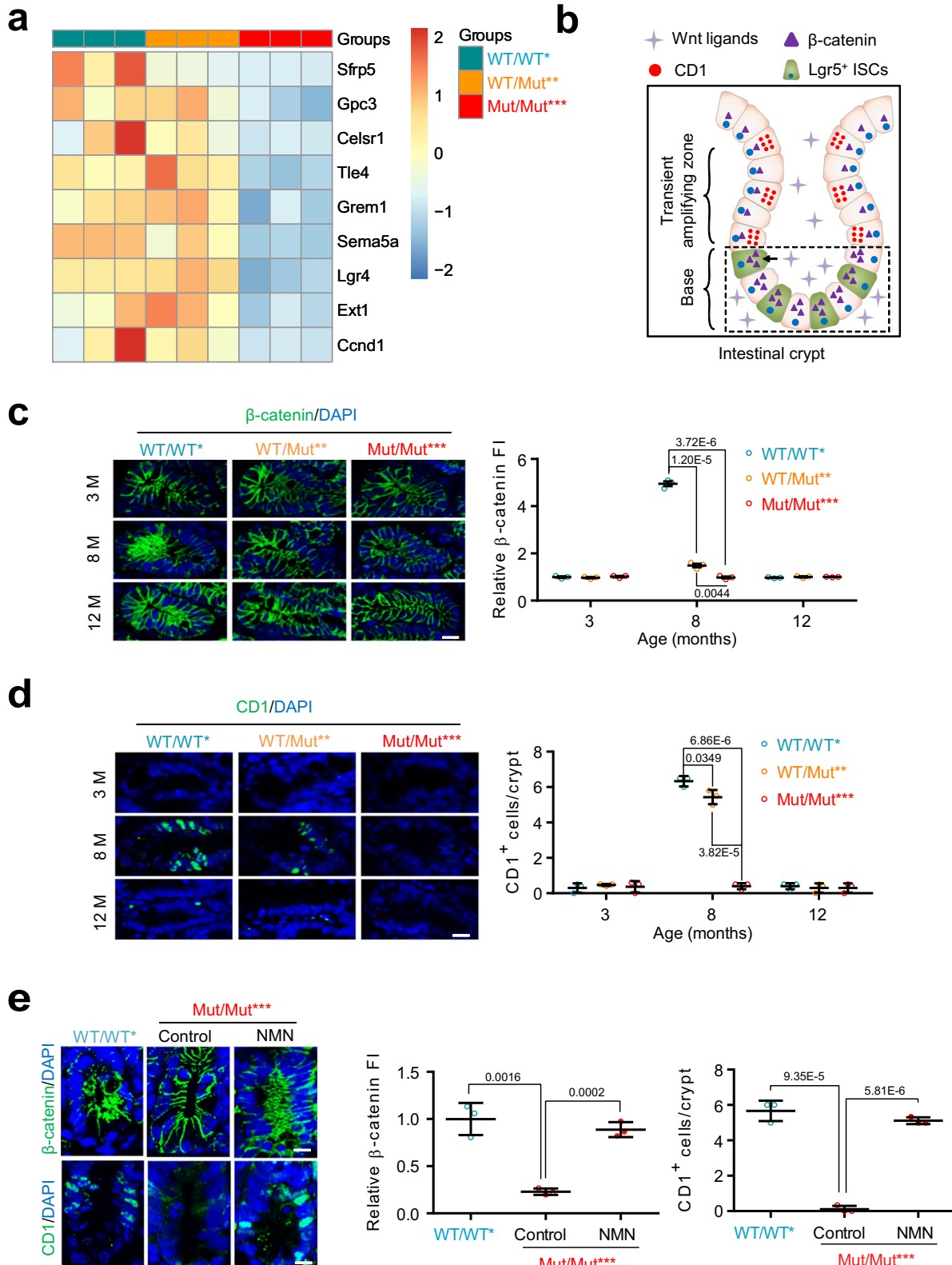

ATF5-dependent UPR^mt activation, which is characterized by LONP1 upregulation.

## Discussion

During the aging process, mtDNA mutations are observed to be accumulated in aged rodent and human tissues[21–23,62–65]. We showed previously that the significant accumulation of much more low-

frequency (less than 0.5%) mtDNA point mutations in human oocytes during aging is linked with impaired blastocyst formation[18]. We also observed the accumulation of low-frequency point mutations in aged mouse intestine, implying an important role in physiological aging. Notably, the aged intestine of POLG mutator mice show a similar but more rapid accumulation of low-frequency mtDNA point mutations versus wild-type mice aged intestine,

**Fig. 4 | NAD⁺ depletion impairs Wnt/β-catenin signaling during the intestinal aging caused by mtDNA mutation burden. a** Heatmap of differential genes associated with Wnt/β-catenin signaling in WT/WT*, WT/Mut** and Mut/Mut*** mice at 8 months of age ($n = 3$ mice per group). **b** Schematic illustration of LGR5-expressing ISCs, proliferated epithelial cells and Wnt/β-catenin signaling in the intestinal crypts. The figure of intestinal crypt was generated with the help of Servier Medical Art, provided by Servier, licensed under a Creative Commons Attribution 4.0 unported license (https://creativecommons.org/licenses/by/4.0/deed.en). **c, d** Anti-β-catenin (**c**) and anti-CD1 (**d**) IF in the intestinal crypts of WT/WT*, WT/Mut** and Mut/Mut*** mice at 3, 8 and 12 months of age. Relative FIs of β-catenin and CD1⁺ cell number per crypt are quantified on right (Scale bar, 10 μm; data are presented as the mean ± S.D and $n = 3$ mice per group; two-way ANOVA test). **e** Anti-β-catenin and anti-CD1 IF in the intestinal crypts of Mut/Mut*** mice at 8 months of age treated with NMN or water. Relative FI of β-catenin and CD1⁺ cell number per crypt are quantified on right (Scale bar, 10 μm; data are presented as the mean ± S.D and $n = 3$ mice per group; unpaired two-tailed Student's $t$-test). Source data are provided with this paper.

---

providing a suitable model for investigating the roles of mtDNA mutation burden on intestinal aging.

Using this mouse model, we found that increased mtDNA mutation burden resulted in NAD⁺ depletion during intestinal aging with subsequent activation of the ATF5-dependent UPRᵐᵗ. These mechanisms regulate mtDNA mutation-induced aging phenotypes and LGR5-positive ISC exhaustion through impaired Wnt/β-catenin signaling (Fig. 7d).

Using in vivo ISC labeling and organoid technology, we defined the landscape of cross-cell signals caused by mtDNA mutation burden in ISCs and niche cells. Wnt/β-catenin signaling is required for ISCs to maintain the balance between proliferation and differentiation[7,66,67]. High levels of Wnt signaling at the base of intestinal crypts activates β-catenin in ISCs, which upregulates its target genes (*c-MYC*, *Cyclin D1*, etc.) to promote ISC differentiation into the transient amplifying (TA) zone[68]. Our study showed that low mtDNA mutation burden activates the Wnt/β-catenin pathway in the base of intestinal crypts and increases the number of CD1-expressing cells in the TA zone during intestinal aging (Fig. 4). Impairment of this cross-cell communication was observed in the aged intestine caused by increased mtDNA mutation burden (Fig. 4), implying an important role in small intestine aging. These findings provide insights into the intestinal cell differentiation processes. Upon aging, intestinal crypt size, villus length, Paneth cell number, and goblet cell number all increase, and the number and regenerative potential of ISCs are reduced upon aging in mice[14,34,69]. It is interesting that intestinal villus length increases while ISC number is decreased in the small intestine, requiring further studies.

NAD⁺ depletion is regarded as a common signature of aging[70,71], which is observed in many tissues during aging[72–75]. It also reportedly compromises mitophagy[41,42], which exacerbates accelerated aging[41,76–79]. Similarly, we observed NAD⁺ depletion and compromised mitophagy during intestinal aging in response to increased mtDNA mutation burden. We also found that NAD⁺ depletion activates the ATF5-dependent UPRᵐᵗ to regulate intestinal aging and that the compromised mitophagy could not be restored by NAD⁺ repletion. Overall, our study provides a mechanism linking NAD⁺ depletion and aging, and supports its utility as a biomarker to monitor the aging of different tissues.

Mitochondrial ISR, triggered by mitochondrial stress, modulates protein synthesis and selectively overexpresses a set of stress-responsive genes to activate pathways including one carbon metabolism, serine acid biosynthesis and antioxidant mechanism/redox homeostasis[48,49,53,54,80–82]. The stress-responsive transcription factors, ATF4[47–50], ATF5[51,52] and/or CHOP[53,54], reportedly mediate the transcription of mitochondrial ISR genes. We found that increased mtDNA mutation burden triggers ISR activation through NAD⁺ depletion. Interestingly, ATF5 but not ATF4 and CHOP is upregulated in the aged Mut/Mut*** intestine, demonstrating a preferentially translation of ATF5.

The UPRᵐᵗ is reportedly activated by various mitochondrial stresses including mitonuclear protein imbalance and mitochondrial import efficiency[60,83], which is marked by increased protein expression of LONP1, HSP60 and ClpP. We found that increased mtDNA mutation burden triggers the ATF5-dependent UPRᵐᵗ activation and resulting LONP1 upregulation, and regulates intestinal aging phenotypes. The specific upregulation of LONP1 protein provides a candidate marker for intestinal aging caused by increased mtDNA mutation burden.

Finally, we showed that NAD⁺ repletion in vitro and in vivo increases Wnt/β-catenin signaling as well as the number of LGR5-expressing ISCs in the small intestine. Moreover, both si*Atf5* and ISRIB enhance the regenerative potential of aged intestinal crypts. These findings identify multiple potential candidates for both treating and preventing intestinal aging.

## Methods
### Mice
*PolgA^D257A* heterozygous mice (*PolgA^WT/Mut*, Stock no: 017341, USA) and *LGR5-EGFP-IRES-creERT2* heterozygous mice (*LGR5^WT/GFP* Stock no: 008875, USA) were purchased from the Jackson Laboratory. The genotypes of siblings were determined by sequencing the genome of mouse tail at 3 weeks old. Wild-type C57BL/6 J mice with different ages (3 months, 8, 12, and 20 months) were purchased from the Gem-Pharmatech co. Ltd (Nanjing, China). *PolgA^WT/Mut* male mice were continuously crossed to WT female mice for maintaining the population of *PolgA^WT/Mut* mice. Then, the male and female *PolgA^WT/Mut* siblings were crossed to generate three genotype mice including WT/WT*, WT/Mut** and Mut/Mut***. Mice were housed in an environment of suitable temperature (25 °C) and humidity (typically 50%) under a 12 h: 12 h light/dark cycle (7 am/7 pm) with accessing to food and water ad libitum. The same batch of male mice was used to perform experiments in accordance with relevant guidelines and regulations, which had been reviewed and approved by Guangzhou Institutes of Biomedicine and Health Ethical Committee (Approve no. 2018040). Mice were euthanized using CO₂ inhalation.

### Histology
Sections of jejunum, the middle region of small intestine, were dissected from mice and fixed in 4% formalin for 24 h and embedded in paraffin. The jejunum sections were mounted on slides and stained with hematoxylin-eosin (H&E), and observed under a light microscope (Zeiss, Germany).

### TUNEL staining
A TUNEL assay kit (In Situ Cell Death Detection Kit, Roche, 45-12156792910) was used to characterize apoptosis in the jejunum sections according to the manufacture's instruction. After being dewaxed and hydrated, the jejunum sections were placed in 200 mL 0.1 M citric acid buffer (pH 6.0) and irradiated with 750 watts microwave for 1 min. The sections were immediately cooled with double distilled water (-25 °C) and washed twice with PBS. After antigen repair, the sections were fixed with 4% paraformaldehyde for 1 h at room temperature, and were subsequently incubated with protease K for 30 min at 37 °C. Then, the slides were immersed in 0.1 M Tris-HCl (pH 7.5) containing 3% bovine serum albumin and 20% normal bovine serum for 30 min at room temperature. The sections were incubated with 50 μL TUNEL reaction mixture in a humid environment under dark for 1 h at 37 °C. After 3 times of PBS washing, the sections were stained with DAPI solution (D9542, Sigma–Aldrich, USA), sealed and visualized under a Zeiss LSM 710 with a 10× objective. The number of TUNEL-positive cells was quantified by ImageJ software.

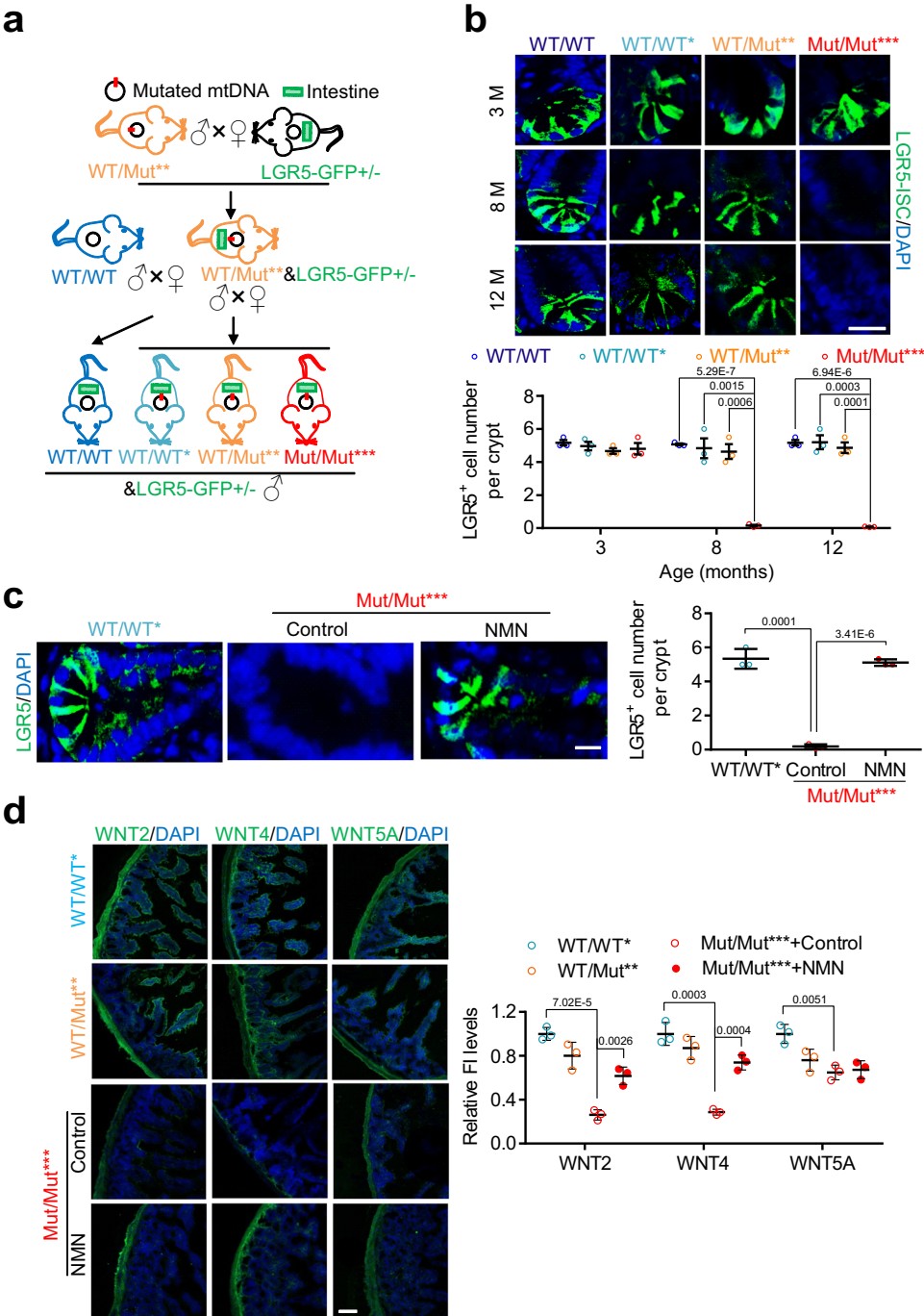

**Fig. 5 | Increased mtDNA mutation burden induces ISC decrease through NAD⁺ depletion. a** Generation of LGR5-GFP reporter mice with four categories of mtDNA mutation burden. *LGR5^{WT/GFP}* female mice were first crossed to WT/Mut** male mice, and the offspring of *LGR5^{WT/GFP}*& WT/Mut** mice were further used to generate WT/WT*&*LGR5^{WT/GFP}*, WT/Mut**&*LGR5^{WT/GFP}* and Mut/Mut***&*LGR5^{WT/GFP}*. WT/WT&*LGR5^{WT/GFP}* mice generated by crossing WT/WT male mice to female *LGR5^{WT/GFP}* female mice were used as an additional control. The figure was partly generated with the help of Servier Medical Art. **b** Representative images of LGR5-expressing ISCs in WT/WT&*LGR5^{WT/GFP}*, WT/WT*&*LGR5^{WT/GFP}*, WT/Mut**&*LGR5^{WT/GFP}* and Mut/Mut***&*LGR5^{WT/GFP}* male mice at 3, 8 and 12 months of age (Scale bar,

20 µm). ISC number is quantified at bottom (Data are presented as the mean ± S.D and *n* = 3 mice per group; two-way ANOVA test). **c** Representative images of LGR5-expressing ISCs in Mut/Mut***&*LGR5^{WT/GFP}* mice at 8 months of age treated with NMN or water control (Scale bar, 10 µm). ISC number is quantified on right (Data are presented as the mean ± S.D and *n* = 3 mice per group; unpaired two-tailed Student's *t*-test). **d** Anti-WNT2, anti-WNT4 and anti-WNT5A IF in the intestine of WT/WT*, WT/Mut** and Mut/Mut*** mice at 8 months of age treated with NMN or water control. Relative FI is quantified on right (Scale bar, 100 µm; data are presented as the mean ± S.D and *n* = 3 mice per group; one-way ANOVA test). Source data are provided with this paper.

## Detection of telomere length

Mouse telomere repeats were measured by real-time quantitative PCR as described[84]. Briefly, total genome of jejunum was prepared using a genome extraction kit (TianGen, China, DP304). The PCR reaction was run on a CFX-96 real-time PCR detection system (BioRad, USA) with an SYBR Green QPCR kit (TaKaRa, RR820A). The primers (forward primer: 5′-CGGTTTGTTTGGGTTTGGGTTTGGGTTTGGGTTTGGGTT-3′, reverse primer: 5′-GGCTTGCCTTACCCTTACCCTTACCCTTACCCTTA

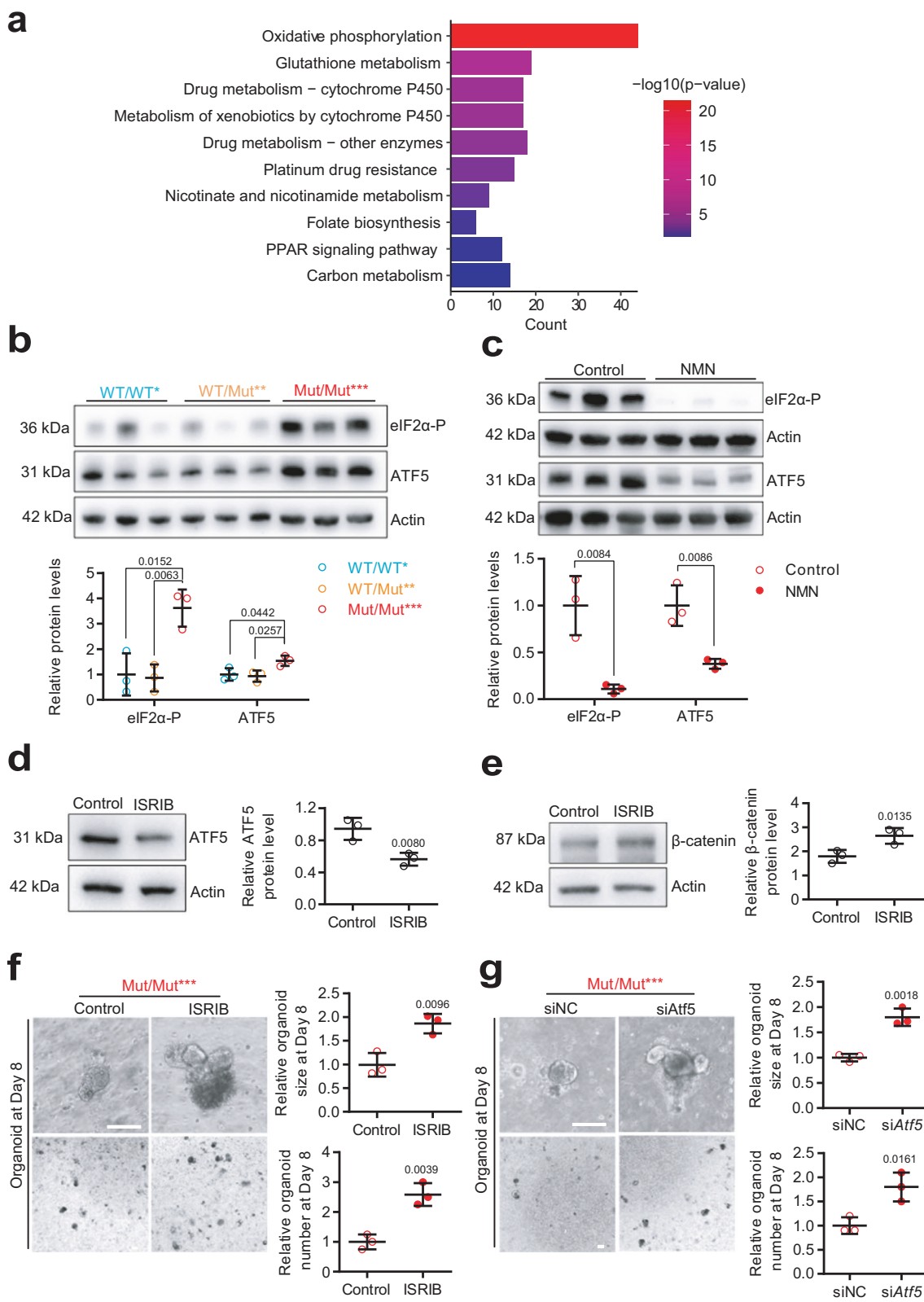

CCCT-3′) were used to detect telomere. Amplification of 36B4 in the same samples was used as an internal control for detecting telomere length, and the primers (forward primer: 5′-ACTGGTCTAGGACCCGA-GAAG-3′, reverse primer: 5′-TCAATGGTGCCTCTGGAGATT-3′) were used to detect 36B4. Relative telomere length was calculated and normalized to values obtained from the amplification of 36B4.

**Analysis of mtDNA mutations**

Total genome of jejunum was prepared using a genome extraction kit (TianGen, China, DP304). Four pairs of primers were used to amplify region 1611-5657, amplified with a forward primer (5′-AAAAGCAGC-CACCAATAAAG-3′) and a reverse primer (5′-GAGAAATGATGGTGG-TAGGA-3′), region 5636-9914, amplified with a forward primer

**Fig. 6 | NAD⁺ depletion activates ISR to regulate mtDNA mutation-induced aging phenotypes through impaired Wnt/β-catenin signaling. a** Kyoto Encyclopedia of Genes and Genomes pathway enrichment of differentially upregulated genes in Mut/Mut*** mice at 8 months of age versus WT/WT* mice was performed using clusterProfiler package, and significance was determined by Fisher exact test of adjusted $p$-value by BH method ($n = 3$ mice per group). **b** Protein expression of eIF2α-P and ATF5 by western blot analysis in the intestinal crypts of WT/WT*, WT/Mut** and Mut/Mut*** mice at 8 months of age. Relative band densities quantified using ImageJ are shown at bottom (Data are presented as the mean ± S.D and $n = 3$ mice per group; one-way ANOVA test). **c** Protein expression of eIF2α-P and ATF5 by western blot analysis in the intestinal crypts of Mut/Mut*** mice treated with NMN or water. Relative band densities quantified using ImageJ are shown at bottom (Data are presented as the mean ± S.D and $n = 3$ mice per group; one-way ANOVA test). **d** Protein expression of ATF5 by western blot analysis in Mut/Mut*** intestinal crypt cells treated with Control or ISRIB. Relative band densities quantified using ImageJ are shown on right (Data are presented as the mean ± S.D and $n = 3$ mice per group;

paired two-tailed Student's $t$-test). **e** Protein expression of β-catenin by western blot analysis in Mut/Mut*** intestinal crypt cells treated with ISRIB or control. Relative band densities quantified using ImageJ are shown on right (Data are presented as the mean ± S.D and $n = 3$ mice per group; paired two-tailed Student's $t$-test). **f** Representative organoid images at Day 8 from the intestinal crypts of Mut/Mut*** mice at 8 months of age with 500 nM ISRIB or water control (Scale bar, 100 μm). Organoid size and relative organoid number per well are quantified on right, and the average organoid number per well is normalized to the control. (Data are presented as the mean ± S.D and $n = 3$ mice per group; unpaired two-tailed Student's $t$-test) **g** Representative organoid images at Day 8 from the intestinal crypts of Mut/Mut*** mice at 8 months of age with si*Atf5* or control siNC (Scale bar, 100 μm); organoid size and relative organoid number per well are quantified at bottom. Average organoid number per well is normalized to the control. Data are presented as the mean ± S.D and $n = 3$ mice per group; unpaired two-tailed Student's $t$-test. Source data are provided with this paper.

---

(5′-ACTCCTACCACCATCATTTC-3′) and a reverse primer (5′-GAGAAG GCTATGGTGAGGTT-3′), region 9876-14278, amplified with a forward primer (5′-TATGCCATCTACCTTCTTCA-3′) and a reverse primer (5′-ATTTGGACTATTAGGCAGAC-3′) and region 14258-1611, amplified with a forward primer (5′-AGTCTGCCTAATAGTCCAAA-3′) and a reverse primer (5′-CTTTATTGGTGGCTGCTTTT-3′), which covers the whole mtDNA sequence.

PCR of mtDNA using the genome of jejunum as template was performed, and the four amplified fragments were mixed for sequencing. Briefly, 3 μg mixture of mtDNA was used to construct a library as described[85] for a better homogeneity of sequencing depth. Then, the library was sent to Berry Genomics. Co., Ltd (Beijing, China) and sequenced on an Illumina Hiseq4000 platform using 150 bp paired end reads for 3 G flux. mtDNA mutations were analyzed as described[15].

### Small intestinal crypt isolation and organoid culture

Small intestinal crypts were isolated from jejunum sections as described[86], and cultured using "ENR" culture medium as described[87,88]. Mice were sacrificed by cervical dislocation. The jejunum was dissected longitudinally, cleaned with PBS at 4 °C, and cut into tissue blocks with 1 centimeter length. After being cleaned 10–20 times with PBS containing 100 U/mL penicillin/streptomycin, the jejunum segments were incubated with 2 mM EDTA solution (pH 8.0) for 20 min at 4 °C, shaken by a vortex shaker for 2 min and centrifuged at 110 g for 5 min. After the supernatant was removed, the intestinal crypts were gently flicked with DMEM/F-12 medium, centrifuged at 110 g for 5 min, and resuspended with ENR medium. 10 μL intestinal crypt suspension and 60 μL matrigel were mixed, added into a 24-well cell culture plate, and polymerized for 15 min at 37 °C. The intestinal crypts were cultured with ENR medium, and the frequency of organoid formation was analyzed as described[35]. siAtf5 (Sense strand sequence: 5′-GCTCGTAGACTATGGGAAAdTdT-3′) was used for silencing the Atf5 genes in the intestinal crypt cells following the procedure described in the Lipofectamine™ RNAiMAX transfection kit. After validating silence efficiency, the siRNA and control siNC (5′-TTCTCCGAACGTGTCACGT-3′) were added into the "ENR" medium. Then, isolated intestinal crypts were cultured with this medium for the first 2 days, and organoid formation was visualized with an optical microscope (ECLIPSE Ts2-FL, Nikon, Japan) at Day 8.

### Detection of mitochondrial inner membrane potential and ROS

For detecting mitochondrial inner membrane potential, the isolated mouse intestinal crypt cells were incubated with JC-1 (Beyotime, C2003S, China) for 30 min at room temperature in the dark. Then, fluorescence intensity was detected using a BD Fortessa flow cytometer (BD Biosciences, USA) within 1 h and analyzed using FlowJo V10 software. The gating strategy is shown in Supplementary Fig. 10. For detecting ROS, isolated mouse intestinal crypt cells were first

seeded on a 20 mm glass-bottom cell culture dish. Then, the cells were incubated with DCF (Invitrogen, D399, USA) for 30 min at room temperature, and imaged using a Zeiss LSM 880 with a 63× objective.

### Immunofluorescence

Immunofluorescence was performed on 6 μm thick paraffin or frozen sections. The paraffin sections of jejunum were deparaffinized, rehydrated, and permeabilized in 10 mM sodium citrate buffer. After being restored to room temperature, the frozen sections of jejunum were first soaked in PBS for 10 min and then blocked for 30 min. Then, all sections were incubated with primary antibodies overnight at 4 °C. After 5 times washing with PBS, the sections were incubated with Alexa Fluor 488 and 568 conjugated secondary antibodies (Life Technologies). Finally, the sections were imaged using a Zeiss LSM 900 or a Zeiss LSM 710 with a 10× objective after incubation with DAPI solution (D9542, Sigma−Aldrich, USA) for 10 min. Antibodies used were as follows: anti-HSP60 (Abcam, ab46798, 1:200), anti-Cyclin D1 (Abcam, ab16663, 1:200), anti-Gli1 (Santa Cruz Biotechnology, sc-515751, 1:100), anti-Foxl1 (Santa Cruz Biotechnology, SC-130373, 1:100), anti-LC3B (Cell Signal Technology, 2775, 1:200), anti-TOM20 (Abcam, ab56783, 1:200), anti-β-catenin (Cell Signal Technology, 9562 S, 1:200), anti-Notch1 (Abclonal, A7636, 1:100), anti-WNT2 (Abclonal, A23997, 1:100), anti-WNT5A (Abclonal, A12744, 1:100), anti-WNT4 (Bio-Techne, MAB4751, 1:100), anti-CDKN1A/p21 (Abclonal, A2691, 1:100), anti-Mouse IgG (H + L), Alexa Fluor 488 (Thermo Fisher Scientific, A-11001, 1:500), anti-Rabbit IgG (H + L), Alexa Fluor 488 (Thermo Fisher Scientific, A-11008,1:500) and anti-Rabbit IgG (H + L), Alexa Fluor 568, (Thermo Fisher Scientific, A-11011, 1:500).

### Imaging for detecting NADH/NAD⁺ ratio and NAD⁺ level

SoNar/cpYFP plasmids for measuring NADH/NAD⁺ ratio, and mCherry-FiNad/mCherry-cpYFP plasmids for measuring the NAD⁺ level were gifts from Professor Yi Yang and Yuzheng Zhao (East China University of Science and Technology, China). The intestinal crypt cells were transfected with these plasmids for 24−48 h, and then seeded on a 20 mm glass-bottom cell culture dish (FluoroDish, World Precision Instruments, FD35-100) for imaging using a Zeiss LSM 710 with a 40× objective. The F405 nm/F488 nm ratios of SoNar and the F488 nm/F543 nm ratios of mCherry-FiNad were calculated as described previously[32–34].

### RNA sequencing

Total RNA of the jejunum sections was extracted using a TRIzol-based protocol. Libraries were prepared following the instructions for the IlluminaTruSeq RNA Sample Prep kit. After sequencing was performed on a Illumina NovaSeq instrument, data were analyzed with RSEM software. Then, DESeq2 was used to identify the differential expressed

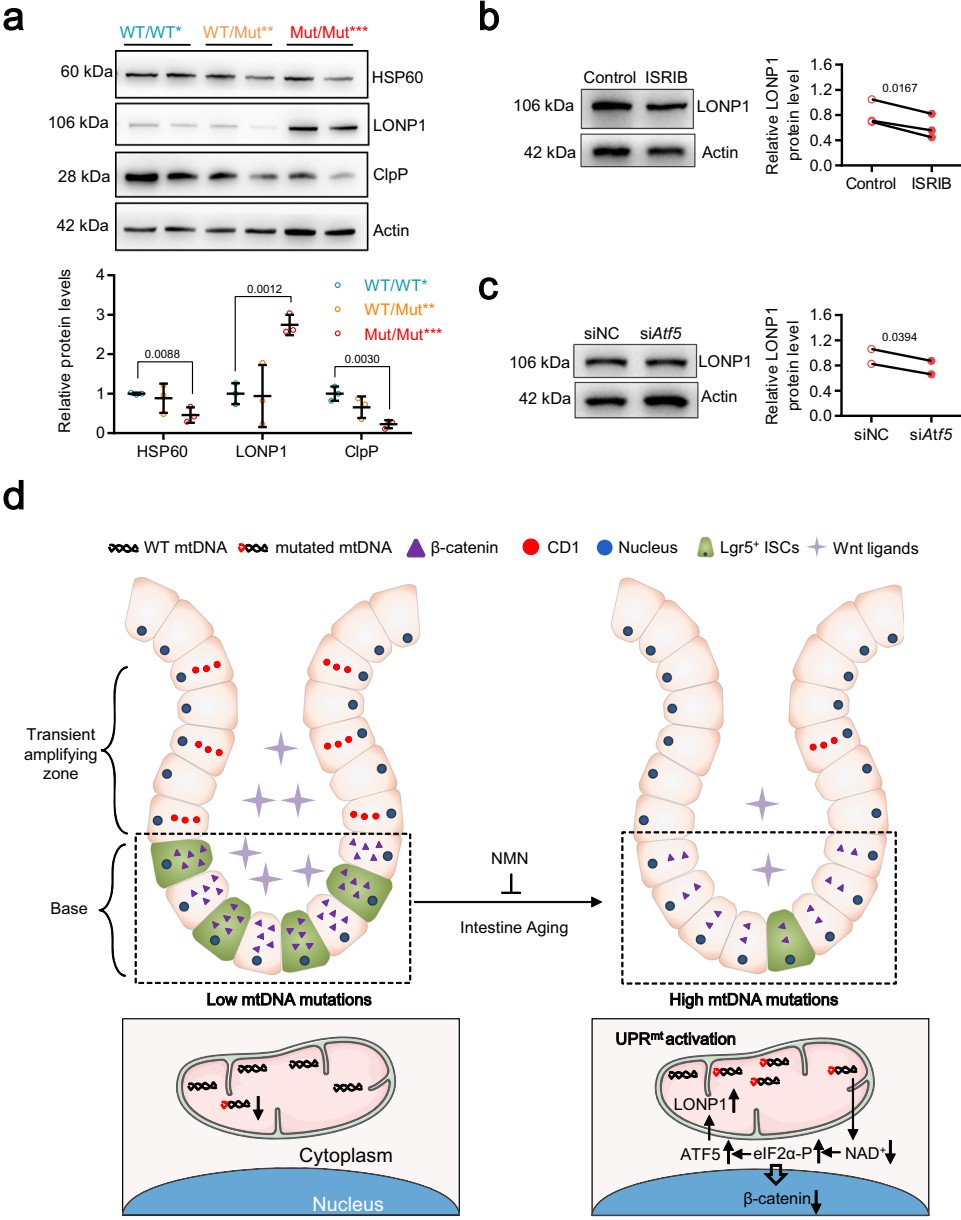

**Fig. 7 | The ISR caused by mtDNA mutation burden regulates UPRmt activation.**
**a** Protein expression of HSP60, LONP1, and ClpP by western blot analysis in the intestinal crypts in WT/WT\*, WT/Mut\*\*, and Mut/Mut\*\*\* mice at 8 months of age. Relative band densities quantified using ImageJ are shown at bottom (Data are presented as the mean ± S.D and $n = 3$ mice per group; one-way ANOVA test). **b** Protein expression of LONP1 by western blot analysis in the intestinal crypts of Mut/Mut\*\*\* mice treated with ISRIB or control. Relative band densities quantified using ImageJ are shown on right (Data are presented as the mean ± S.D and $n = 3$ mice per group; paired two-tailed Student's $t$-test). **c** Protein expression of LONP1 by western blot analysis in the intestinal crypts of Mut/Mut\*\*\* mice treated with si*Atf5* or siNC. Relative band densities quantified using ImageJ are shown on right (Data are presented as the mean ± S.D and $n = 2$ mice per group; paired two-tailed Student's $t$-test). **d** Summary of mechanism regulating intestinal aging caused by mtDNA mutation burden. The figures of small intestine and mitochondria were generated with the help of Servier Medical Art, provided by Servier, licensed under a Creative Commons Attribution 4.0 unported license (https://creativecommons.org/licenses/by/4.0/deed.en).

genes, with $p$-adjust <0.05 & abs (log$_2$FoldChange)>0.5. Gene Ontology was performed using ClusterProfiler.

## NMN and ISRIB supplementation

For NAD$^+$ repletion in vivo, β-NMN (1094-61-7, Sigma–Aldrich, USA) was administered in drinking water to treat mice for 2 weeks as described[53]. For NAD$^+$ repletion in vitro, β-NMN solution was first prepared into autoclaved water at the certain dose and sterilely filtered, and then added into "ENR" medium for culturing intestinal crypt cells. For inhibiting ISR in vitro, 500 nM ISRIB (S0706, Selleck, USA)

was first prepared into autoclaved water at the certain dose and filtering sterilely as described[89], and then added into "ENR" medium for culturing intestinal crypt cells.

## SIRT7 overexpression

The pLVX-SIRT7-Puro and control pLVX-Puro plasmids were separately transfected into intestinal crypt cells following the instructions for the Lipofectaminew 3000 Reagent kit (L3000015, Invitrogen, USA). After 48 h, intestinal crypt cells were lysed, and then protein expression of SIRT7 and eIF2α-P was assayed by western blotting.

## Measurement of NAD⁺ content

Small intestinal crypts were isolated and sent to Tsinghua University (Beijing, China) for determining $NAD^+$ content using a liquid chromatography-tandem mass spectrometry (LC–MS/MS). The ACQUITY UPLC H-Class system was coupled a 6500plus QTrap mass spectrometer (AB SCIEX, USA), equipped with a heated electrospray ionization (HESI) probe. Extracts were separated by a synergi Hydro-RP column (2.0 × 100 mm, 2.5 μm, phenomenex). The mobile phase consisted of a binary solvent system: mobile phase A (2 mM triisobutylamine adjusted with 5 mM acetic acid in water) and mobile phase B (methanol). A 15-min gradient with flow rate of 250 μL/min was used as follows: 0–1.5 min, 5%B; 1.5–9 min, 5–35% B; 9.5–12 min, 98% B;12.1–15 min, 5%B. Column chamber and sample tray were held at 35 °C and 10 °C, respectively, and data were acquired in multiple reaction monitor (MRM) mode. The nebulizer gas (Gas1), heater gas (Gas2), and curtain gas were set at 50, 50, and 35 psi, respectively. The ion spray voltage was −4500 V in negative ion mode. The optimal probe temperature was determined to be 450 °C. The SCIEX OS 1.6 software was applied for metabolite identification and peak integration.

## Western blotting

Equal amounts of total protein extracted from the intestinal crypt cells, were electrophoresed on 12% polyacrylamide/sodium dodecyl sulfate gel and transferred onto polyvinylidene fluoride membranes. After being blocked for 45 min, the membranes were incubated with primary antibodies for 1 h. Subsequently, membranes were incubated with horseradish peroxidase-coupled secondary antibodies for 1 h after 3 times washing with PBS. Finally, the immunoreactivity was detected using Immobilon Western Chemiluminescent HRP Substrate (Millipore, USA, WBKLS0500). The antibodies used were as follows: anti-HSP60 (Abcam, ab46798, 1:1000), anti-LONP1 (Cell Signal Technology, 28020, 1:1000), anti-CDKN1A/p21 (Abclonal, A2691, 1:100), anti-Phospho-eIF2α-S51 (Abclonal, AP0692, 1:1000), anti-ClpP (Abcam, ab124822, 1:1000), anti-ATF4 (Abcam, ab216839, 1:1000), anti-ATF5(Abcam, ab184923, 1:1000), anti-CHOP(Cell Signal Technology, 2895 S, 1:1000), anti-TOM20 (Proteintech, 11802-1-AP, 1:1000), anti-WNT4 (Bio-Techne, MAB4751, 1:1000), anti-SIRT7 (Abclonal, A22735, 1:1000), anti-Actin (HUABIO, ET1702-67, 1:3000), HRP Conjugated Goat anti-Mouse IgG (HUABIO, HA1006, 1:3000) and HRP Conjugated Goat anti-Rabbit IgG (HUABIO, HA1001,1:3000).

## Statistical analysis

Data are shown as mean ± Standard Deviation (SD). All statistical analysis was performed using Student's $t$-test or ANOVA test from the GraphPad Prism software (GraphPad) as indicated in the figure legends. $P$ values of less than 0.05 were considered as significant.

## Reporting summary

Further information on research design is available in the Nature Portfolio Reporting Summary linked to this article.

## Data availability

The raw RNA-seq and mtDNA-seq data have been deposited in the Genome Sequence Archive (GSA) at the Beijing Institute of Genomics (BIG) Data Center, BIG, Chinese Academy of Sciences (https://bigd.big.ac.cn/gsa), and the accession numbers are CRA012233 for RNA-seq (https://ngdc.cncb.ac.cn/gsa/browse/CRA012233) and CRA012237 for mtDNA-seq (https://ngdc.cncb.ac.cn/gsa/browse/CRA012237) that are publicly accessible. The LC–MS/MS data for measuring $NAD^+$ content has been deposited in the MassIVE data repository (https://massive.ucsd.edu/) and are available with the accession code MSV000093751 (https://massive.ucsd.edu/ProteoSAFe/dataset.jsp?accession=MSV000093751). All other relevant data supporting the key findings of this study are available within the article and its supplementary information files. Any

additional information is available upon request to the corresponding author (Xingguo Liu, liu_xingguo@gibh.ac.cn). Source data are provided with this paper.

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

## Acknowledgements

This work was financially supported by the National Key Research and Development Program of China (2022YFA1103800), the National Natural Science Foundation projects of China (32025010, 92157202, 32241002, 92254301, 32261160376, 31970709, 32070729, 32100619, 32170747, 32322022, 32370782, 32371007, 32300608, 32300620, 92357302), NSFC/RGC Joint Grant Scheme 2022/2023 (N_CUHK 428/22), the Stra-tegic Priority Research Program of the Chinese Academy of Sciences (XDB0480000), the National Key Research and Development Program of China (2022YFE0210100, 2019YFA0904500, 2023YFE0210100), the Key Research Program, CAS (ZDBS-ZRKJZ-TLC003), International Cooperation Program, CAS (154144KYSB20200006), CAS Project for Young Scientists in Basic Research (YSBR-075), Guangdong Province Science and Technology Program (2023B1111050005, 2023A1515030231, 2022A1515110493, 2023B1212060050, 2021A1515012513, 2021B1515020096, 2022A1515012616, 2022A1515110951), Guangzhou Science and Technology Program (202102021037, 202102020827, 202102080066, 202206060002, 2023A04J0414), Health@InnoHK funding support from the Innovation Technology Commission of the Hong Kong SAR and CAS Youth Inno-vation Promotion Association (to Y. W and K. C). We thank the Experi-mental Animal Center and Public Instrument Center in Guangzhou Institutes of Biomedicine and Health, and the Metabolomics Facility Center of Metabolomics and Lipidomics in National Protein Science Technology Center of Tsinghua University for LC–MS/MS experiments.

## Author contributions

X. Liu initiated and supervised the project. L.Y., Z.R., and X. Lin designed, performed the experiments, and analyzed the data. H.W., Y.X., H.T., Z.H., J.W., Y.Z., and Y.W. participated in the experiments. D.Q., G.L., K.M.L., and W-Y.C. gave suggestions. X. Liu, L.Y., and Z.R. wrote the manuscript.

## Competing interests

The authors declare no competing interests.
