## [Peer Review File · Nature Communications]

NAD⁺ dependent UPRmt activation underlies intestinal aging caused by mitochondrial DNA mutationsReviewers' comments:

Reviewer #1 (Remarks to the Author):

The authors have used mtDNA mutator mice that carry an error prone version of POLG, the only DNA polymerase that replicates mtDNA, hence the mice accumulate increasing levels of mtDNA mutations over time. Using this model, the authors have analyzed the effect of mtDNA mutations on intestinal cells, crypts and stem cells, mainly using organoid technology and intestinal cell labelling. The authors suggested that low and high rates of mtDNA mutations have very different effects on the small intestine and mechanistically try to connect this to NAD⁺ depletion and LONP1- dependent UPR_{mt} inhibition. Ultimately, they show that NMN treatment can partially reverse the phenotypes, although it is not clear the physiological outcome of this intervention. Although the overall results are interesting, there are quite a few issues with the interpretation of the results and the technical approaches used. Furthermore, the mechanism of how NAD⁺ - LONP1 – HSP60 axis works and how is connected with high levels of mtDNA mutations is missing.

1. The major problem of this manuscript is the way of how mtDNA mutation levels are determined. PCR amplification of different regions combined with sequencing is extremely prone to mistakes and therefore should not be used to estimate the number of mtDNA mutations. This has been shown repeatedly and state-of-the-art is next generation sequencing of freshly isolated mtDNA from purified mitochondria, without further manipulation by PCR. PCR amplification not only introduces a bias to the study of mtDNA, it also makes it difficult to identify rearrangements of mtDNA such as deletions (linear deletions are major phenotype of mtDNA mutator mice) and duplications unless full-length amplification of mtDNA is performed.

2. The authors introduce mice with different levels of mtDNA mutations in Figure 1 and then only in Fig 2 measure the actual mtDNA levels. There is not clear explanation of how is the breeding scheme carried out and how often have animals been intercrossed in this way. This has to be added to Material and Methods. Previously, it has been shown that POLGmut heterozygous mothers would leak a certain number of mtDNA mutations to its progeny, even wild type ones. However, this can only be observed after certain number of intercrossing, even when Next Generation DNA sequencing of highly purified mtDNA is used (Isokallio and Stewart, 2021). Therefore, it is not clear why would WT and WT/mut offspring of POLGmut heterozygous mothers accumulate additional mutations. Furthermore, the difference in the mutation load between naïve WT (offspring of WT mothers) and WT* and WT/mut** (offspring of WT/mut** mothers) is in the level of mtDNA insertions, that are by far the least abundant mtDNA mutation types introduced by error prone POLG (Dayama et al., 2014) in mtDNA mutator mice. How would these small differences in the number of insertions affect mitochondria is not clear. As each cell has 1000 -10.000 copies of mtDNA molecules, not only the general number of mtDNA mutations have to be very high, but individual mutation has to be present in vast majority of mtDNA molecules (threshold is usually over 85% for point mutations) to cause phenotypes. Therefore, the differences reported here between WT, WT* and WT/mut** cannot be explained by the observed (minor) differences in mtDNA insertions. Therefore, the results of mitophagy increase in the WT* and WT/mut** are at least puzzling.

Minor comments on the same note:

a) The authors introduce mice with different levels of mtDNA mutations in Figure 1 and then only in Fig 2 measure the actual mtDNA levels.

b) There is not clear explanation of how is the breeding scheme carried out and how often have animals been intercrossed in this way. This has to be added to Material and Methods as is important for the interpretation of the results. Previously it was shown that extensive interbreeding causes appearance of additional phenotypes that are not directly connected with mtDNA mutation load (Ross et al., 2013).

c) The types of mtDNA mutations, deletions and insertions have to be reported. This is very important for the quality control of the data, as many mtDNA mutator mutation spectra are published over the years.

3. MtUPR has been highly disputed as a major stress response to respiratory chain dysfunction in mammals. In *C. elegans* where mtUPR has been intensely studied HSP60 is claimed to be highly upregulated, hence used as a marker of mtUPR. In mammals, multiple studies over last 5-10 years have described mitochondrial Integrated Stress Response (mitoISR) as a mechanism by which mitochondria signal OXPHOS deficiency to the nucleus (Bao et al., 2016; Fessler et al., 2020; Guo et al., 2020; Kaspar et al., 2021; Khan et al., 2017; Kuhl et al., 2017; Mick et al., 2020; Quiros et al., 2017). This stress response is dependent on the ATF4 expression that remodels metabolism (primary toward 1C/folate metabolism) and induces the upregulation of multiple proteins, including LONP1. The authors should analyze the RNASeq data for signatures of ATF4 targets and ISR. How these data are presented in the manuscript right now, really resembles cherry picking.
4. OXPHOS deficiency in mtDNA mutator mice will cause a strong decrease in NAD⁺/NADH levels. Supplementation by NAD⁺ precursors to alleviate phenotypes of OXPHOS deficiency is not a novel concept, although it is nice to see that it works also in this system. The mechanistic connection to LONP1 – HSP60 provided in this manuscript is at best indirect.
5. Results on the figure 7 are really baffling. First of all, HSP60 is by far the most abundant protein in mitochondria. Therefore, it is not clear, why it suddenly disappears from 12M old samples (immunofluorescence). Are mitochondria in all genotypes suddenly gone? The same goes for the immunohistochemistry upon NMN treatment. What about naïve WT samples of the same ages? The authors should at least provide western blot of HSP60 and LONP1 at all ages and all genotypes. Including naïve WT control. Actually, it would be very helpful to compare proteome of all different samples and ages for the observed changes. Additional western blots used for the quantification should be provided.
6. The very last result panel of figure 7 (7D) is even more puzzling. The depletion of LONP1 provides rescue for the organoids in culture! Mutations in LONP1 cause several devastating diseases in patients and the loss of LONP1 in mice causes very early embryonic lethality. LONP1 has been shown to be THE major mitochondrial protease important for the removal of unfolded and misfolded proteins. Therefore, it is not clear why would its removal improve phenotypes in the mtDNA mutator model.
7. The manuscript does not address the function of mitochondria in the investigated models. At least some data on the level and function of OXPHOS should be provided.

Bao, X.R., Ong, S.E., Goldberger, O., Peng, J., Sharma, R., Thompson, D.A., Vafai, S.B., Cox, A.G., Marutani, E., Ichinose, F., et al. (2016). Mitochondrial dysfunction remodels one-carbon metabolism in human cells. *Elife* 5. 10.7554/eLife.10575.

Dayama, G., Emery, S.B., Kidd, J.M., and Mills, R.E. (2014). The genomic landscape of polymorphic human nuclear mitochondrial insertions. *Nucleic Acids Res* 42, 12640-12649. 10.1093/nar/gku1038.

Fessler, E., Eckl, E.M., Schmitt, S., Mancilla, I.A., Meyer-Bender, M.F., Hanf, M., Philippou-Massier, J., Krebs, S., Zischka, H., and Jae, L.T. (2020). A pathway coordinated by DELE1 relays mitochondrial stress to the cytosol. *Nature* 579, 433-437. 10.1038/s41586-020-2076-4.

Guo, X., Aviles, G., Liu, Y., Tian, R., Unger, B.A., Lin, Y.T., Wiita, A.P., Xu, K., Correia, M.A., and Kampmann, M. (2020). Mitochondrial stress is relayed to the cytosol by an OMA1-DELE1-HRI pathway. *Nature* 579, 427-432. 10.1038/s41586-020-2078-2.

Isokallio, M.A., and Stewart, J.B. (2021). High-Throughput Detection of mtDNA Mutations Leading to tRNA Processing Errors. *Methods Mol Biol* 2192, 117-132. 10.1007/978-1-0716-0834-0_10.

Kaspar, S., Oertlin, C., Szczepanowska, K., Kukat, A., Senft, K., Lucas, C., Brodesser, S., Hatzoglou, M., Larsson, O., Topisirovic, I., and Trifunovic, A. (2021). Adaptation to mitochondrial stress requires CHOP-directed tuning of ISR. *Sci Adv* 7. 10.1126/sciadv.abf0971.

Khan, N.A., Nikkanen, J., Yatsuga, S., Jackson, C., Wang, L., Pradhan, S., Kivela, R., Pessia, A., Velagapudi, V., and Suomalainen, A. (2017). mTORC1 Regulates Mitochondrial Integrated Stress Response and Mitochondrial Myopathy Progression. *Cell Metab* 26, 419-428 e415. 10.1016/j.cmet.2017.07.007.

Kuhl, I., Miranda, M., Atanassov, I., Kuznetsova, I., Hinze, Y., Mourier, A., Filipovska, A., and Larsson, N.G. (2017). Transcriptomic and proteomic landscape of mitochondrial dysfunction reveals secondary coenzyme Q deficiency in mammals. *Elife* 6. 10.7554/eLife.30952.

Mick, E., Titov, D.V., Skinner, O.S., Sharma, R., Jourdain, A.A., and Mootha, V.K. (2020). Distinct mitochondrial defects trigger the integrated stress response depending on the metabolic state of the cell. *Elife* 9. 10.7554/eLife.49178.

Quiros, P.M., Prado, M.A., Zamboni, N., D'Amico, D., Williams, R.W., Finley, D., Gygi, S.P., and Auwerx, J. (2017). Multi-omics analysis identifies ATF4 as a key regulator of the mitochondrial stress response in mammals. *J Cell Biol* 216, 2027-2045. 10.1083/jcb.201702058.

Ross, J.M., Stewart, J.B., Hagstrom, E., Brene, S., Mourier, A., Coppotelli, G., Freyer, C., Lagouge, M., Hoffer, B.J., Olson, L., and Larsson, N.G. (2013). Germline mitochondrial DNA mutations aggravate ageing and can impair brain development. *Nature* 501, 412-415. 10.1038/nature12474.

Reviewer #2 (Remarks to the Author):

Yang et al show that mitochondrial mutator mice have increased villus length and reduced basal crypt number, reduced ISC number, defective b-catenin activation, reduced NAD. NAD depletion partially rescues the defects. The mutant mice failed to upregulate mitophagy at mid age independent of NAD. HSP60 increases with aging, consistent with increased mitochondrial protein folding stress with aging. However, HSP60 is reduced in mutant mice and can be rescued by NAD depletion. Mutant mice have increased LONP1 instead. LONP1 knockdown increases HSP60 and rescues the intestine defects in vitro.

Fig 1b. There is little difference in villus length and basal crypt number in WT mice of various ages. It is unclear whether these are indeed aging related changes based on these data, raising the question of the physiological significance of the changes in mitochondrial mutator mice.

Fig 2a. There is no difference in point mutation, insertion and deletion in WT mice of various ages. These data do not support increased mtDNA mutation with age, again raising the question of the physiological significance of the changes in mitochondrial mutator mice.

Fig 7e. The authors propose that low mtDNA mutations result in mitophagy. There is no evidence to support this. Fig 5a show increased mitophagy in WT mice at 8 months old. But these mice do not have increased mtDNA mutation (Fig 2) compared WT mice at 3 months old. The authors propose mitophagy occurs upstream of NAD depletion and prevents mtDNA mutation and aging. But there is no evidence to support it. To support this claim, the authors need to activate mitophagy to rescue these defects and inactivate mitophagy to speed up these defects.

Another major issue is the mechanism. The authors propose high levels of mitochondrial mutation results in NAD depletion, which causes increased LONP1, and in turn causes decreased HSP60. First, how does NAD depletion cause increased LONP1? Is LONP1 expression dependent on NAD? Second, there seems to be some confusion in the concept of mitochondrial unfolded protein response. Mitochondrial unfolded protein response includes the transcriptional induction of mitochondrial chaperones such as HSP60 and proteases such as LONP1 and ClpP.

Fig 7d. It is hard to imagine without the protection of LONP1, it will be beneficial to the cells or the tissues. Has the authors done the rescue experiment in vivo?

Reviewer #3 (Remarks to the Author):

Mitochondrial dysfunction plays an important role in cell senescence and organism aging. In the manuscript, Yang et al identified an AD+-LONP1-UPRmt pathway that underlies intestinal aging caused by mitochondrial DNA mutations. Using polymerase gamma (POLG) mutator mice, the authors constructed an intestinal aging model and tried to elucidate the underlying mechanisms by which mtDNA mutations causes aging of LGR5-expressing intestinal stem cells. They found that 1) low rates of mtDNA mutation activate mitophagy to curb the mutational load; 2) high levels of mtDNA mutations result in NAD+ depletion and altered LONP1-dependent UPRmt pathway; 3) the intestinal aging phenotype can be reversed by supplementation with NAD+ precursor, NMN; 4) ISC aging is accompanied by a decrease in Wnt-beta-catenin activation; 5) Lonp1 silencing can reverse the aging phenotypes. Overall, the manuscript provides some valuable information regarding ISC aging, however, it is premature to publish these results in Nat Commun at this stage.

Major concerns:

1. In the aging model, at the late stages, the authors show that ISCs are almost gone and the enterocytes undergo extensive apoptosis. Intriguingly, the authors also show there is an increase in the size of the villi. How could this happen?
2. The authors show a decrease in beta-catenin activation in the aging model, which is likely the most important driving force for ISC renewal. Wnt molecules are believed to be supplied by the niche of ISCs: the Gli1+ or Foxl1+ stromal cells or Paneth cells. Do mitochondrial DNA mutations affect the niche cells or the secretion of Wnt molecules by the niche cells?
3. The link between NAD depletion and the Wnt-Catenin pathway and the link between NAD depletion and UPR need to be established. These findings appear to be fragmented at this stage.
4. It will be nice if the authors can provide some genetic data to verify the pathway underlying mtDNA mutations-induced ISC aging.

Minor points:

Both the Introduction and Discussion can be expanded.

Reviewer #1 (Remarks to the Author):

The authors have used mtDNA mutator mice that carry an error prone version of POLG, the only DNA polymerase that replicates mtDNA, hence the mice accumulate increasing levels of mtDNA mutations over time. Using this model, the authors have analyzed the effect of mtDNA mutations on intestinal cells, crypts and stem cells, mainly using organoid technology and intestinal cell labelling. The authors suggested that low and high rates of mtDNA mutations have very different effects on the small intestine and mechanistically try to connect this to NAD⁺ depletion and LONP1-dependent UPR_{mt} inhibition. Ultimately, they show that NMN treatment can partially reverse the phenotypes, although it is not clear the physiological outcome of this intervention. Although the overall results are interesting, there are quite a few issues with the interpretation of the results and the technical approaches used. Furthermore, the mechanism of how NAD⁺–LONP1–HSP60 axis works and how is connected with high levels of mtDNA mutations is missing.

1. The major problem of this manuscript is the way of how mtDNA mutation levels are determined. PCR amplification of different regions combined with sequencing is extremely prone to mistakes and therefore should not be used to estimate the number of mtDNA mutations. This has been shown repeatedly and state-of-the-art is next generation sequencing of freshly isolated mtDNA from purified mitochondria, without further manipulation by PCR. PCR amplification not only introduces a bias to the study of mtDNA, it also makes it difficult to identify rearrangements of mtDNA such as deletions (linear deletions are major phenotype of mtDNA mutator mice) and duplications unless full-length amplification of mtDNA is performed.

Response: We appreciate this comment, and have rewritten our methods more concretely for mtDNA sequencing as suggested. Considering the concerns about the mtDNA sequencing methods, we moved all mtDNA sequencing results as supplementary data, including mtDNA mutations in the intestine, liver and brain of POLG mutator mice (previous Fig. 2 moved into Fig. S2, and new added Fig. S3), and mtDNA mutations in the physiological aged small intestine of 20 month-old wild-type mice (new added Fig. S1).

Actually, we have tried to isolate purified mitochondria using Qproteome Mitochondria Isolation Kit (37612) and other classic methods, but extracted mtDNA mixed with a large amount of nuclear genome. For the existence of nuclear mitochondrial pseudogenes (Numts), the extracted unpurified mtDNA are not chosen for next generation sequencing.

Then, we have tried full-length amplification of mtDNA and two overlap amplification of mtDNA with the same primers, but couldn't amplified all the mtDNA samples until we used 4 overlap amplification of mtDNA. To reduce the mistake by PCR amplification for next generation sequencing, we constructed

a PCR-free library without further amplification of the mtDNA, and indeed have acquired a better homogeneity of sequencing depth. Finally, we had to choose the 4 overlap amplification of mtDNA with a PCR-free library for sequencing the mtDNA samples. The question for sequencing mtDNA more precisely, as you mentioned, is what our lab want to solve for a long time.

As in text:

PCR of mtDNA using the genome of jejunum as template was performed, and the four amplified fragments with an approximate equal molar quantity were mixed and sent to HeQin Bio-Technology Co., Ltd (Guangzhou, China) for sequencing. In briefly, 3 μ g mixture of mtDNA was used to construct a PCR-free library without amplification of the mtDNA for a better homogeneity of sequencing depth. Then, the library was sequenced on an Illumina NovaSeq 6000 platform using 150bp paired end reads for 3G flux. mtDNA mutations were analyzed as described.

As well as aged human clinical intestinal samples 21-23, small intestine of aged mouse intestine, marked by higher expression of CDKN1A/p21 (a well-known senescence marker) and shorter telomere length (Supplementary Fig.1a, b), also accumulates more mtDNA mutations, primarily low-frequency (less than 0.05) point mutations (Supplementary Fig.1c, d), demonstrating a link between mtDNA mutations and the physiological intestinal aging.

As compared to WT/WT mice, at 8 months Mut/Mut^{***} mice accumulate much more of point mutations, primarily low-frequency (less than 0.05) point mutations which is also observed in the small intestine of aged wild-type mice (Supplementary Fig. 2a-d).

Due to the heterozygous mutation of Polg gene in WT/Mut^{**} mice, the brain and liver did accumulate much more expected mtDNA point mutations but small intestine didn't, implying a possible mechanism for clearing mutated mtDNAs in the small intestine (Supplementary Fig. 2b-d and Supplementary Fig. 3a, b).

2. The authors introduce mice with different levels of mtDNA mutations in Figure 1 and then only in Fig 2 measure the actual mtDNA levels. There is not clear explanation of how is the breeding scheme carried out and how often have animals been intercrossed in this way. This has to be added to Material and Methods. Previously, it has been shown that POLGmut heterozygous mothers would leak a certain number of mtDNA mutations to its progeny, even wild type ones. However, this can only be observed after certain number of intercrossing, even when Next Generation DNA sequencing of highly purified mtDNA is used (Isokallio and Stewart, 2021). Therefore, it is not clear why would WT and WT/mut offspring of POLGmut heterozygous mothers

accumulate additional mutations. Furthermore, the difference in the mutation load between naïve WT (offspring of WT mothers) and WT* and WT/mut** (offspring of WT/mut** mothers) is in the level of mtDNA insertions, that are by far the least abundant mtDNA mutation types introduced by error prone POLG (Dayama et al., 2014) in mtDNA mutator mice. How would these small differences in the number of insertions affect mitochondria is not clear. As each cell has 1000 -10.000 copies of mtDNA molecules, not only the general number of mtDNA mutations have to be very high, but individual mutation has to be present in vast majority of mtDNA molecules (threshold is usually over 85% for point mutations) to cause phenotypes. Therefore, the differences reported here between WT, WT* and WT/mut** cannot be explained by the observed (minor) differences in mtDNA insertions. Therefore, the results of mitophagy increase in the WT* and WT/mut** are at least puzzling.

Response: We appreciate this comment, have added clear explanation of the breeding scheme to Material and Methods, have added additional mtDNA sequencing of liver and brain in WT/WT*, WT/Mut** and Mut/Mut*** mice, have revised the presentation about mitophagy, and have moved previous Figure 5 about mitophagy into supplementary data (Fig. S6) as suggested.

For harboring the heterozygous mutation of POLGA, POLGmut heterozygous mothers would theoretically accumulate more mtDNA mutations during aging process, which is observed in POLGmut heterozygous liver and brain at 8 months (new added Fig. S3). The theoretically accumulated mtDNA mutations in POLGmut heterozygous mother at the breeding age would leak a certain number of mtDNA mutations to its progeny due to the maternal inheritance of mtDNA. Thus, we design the breeding pairs to generate mice with four theoretically increasing levels of mtDNA mutations. Indeed as you mentions that the progeny didn't accumulate mtDNA mutations unless after certain number of intercrossing, we also didn't observe the mtDNA mutation difference between naive WT and WT*. So we changed our presentation as suggested, and described WT/WT, WT/WT*, WT/Mut** and Mut/Mut*** as mice with four theoretically increasing levels of mtDNA mutations.

For harboring the heterozygous mutation of POLGA, WT/Mut** mice accumulate much more mtDNA point mutations in liver and brain at 8 months as expected. But the small intestine of WT/Mut** mice didn't accumulate more mtDNA point mutations at 8 months, indicating a possible clearance of mtDNA mutations by mitophagy activation which was observed in the small intestine of WT/WT* and WT/Mut** at 8 months. The mitophagy activation but not the least abundant mtDNA insertion mutations in WT/Mut*** mice provides a possible explanation for the observed difference. We revised all the inappropriate and un-logical presentation about mitophagy in our new version of manuscript.

As in text:

As compared to WT/WT mice, at 8 months Mut/Mut^{***} mice accumulate many more of insertion, deletion, and point mutations, especially low-frequency (less than 0.05) point mutations which is also observed in aged wild-type mice (Supplementary Fig. 2a-d). Due to the heterozygous mutation of Polg gene in WT/Mut^{**} mice, the brain and liver did accumulate much more expected mtDNA point mutations but small intestine didn't, implying a possible mechanism for clearing mutated mtDNAs in the small intestine (Supplementary Fig. 2b-d and Supplementary Fig. 3a, b).

As NAD⁺ depletion has been reported to compromise mitophagy^{40,41}, it raises a possibility that NAD⁺ repletion could activate mitophagy to remove mutated mtDNAs and thus alleviates the aging phenotypes of small intestine caused by mtDNA mutations. By detecting mitophagy using anti-LC3B and anti-TOM20 IF in the intestine crypts of WT/WT^{*}, WT/Mut^{**} and Mut/Mut^{***} mice at 3, 8 and 12 months, we indeed observed compromised mitophagy in Mut/Mut^{***} mice at 8 months (Supplementary Fig. 6a). However, NAD⁺ repletion *in vivo* doesn't rescue mitophagy deficiency or mtDNA mutations in the intestinal crypts of Mut/Mut^{***} mice at 8 months (Supplementary Fig. 6b, c), indicating that the intestinal aging induced by NAD⁺ depletion is independent of mitophagy.

PolgA^{WT/Mut} male mice were continuously crossed to WT female mice for maintaining the population of PolgA^{WT/Mut} mice. Then, the male and female PolgA^{WT/Mut} siblings were crossed to generate three genotype mice including WT/WT^{*}, WT/Mut^{**} and Mut/Mut^{***}. The same batch of male mice was used to perform experiments in accordance with relevant guidelines and regulations, which had been reviewed and approved by Guangzhou Institutes of Biomedicine and Health Ethical Committee (Approve no. 2018040).

Minor comments on the same note:

a) The authors introduce mice with different levels of mtDNA mutations in Figure 1 and then only in Fig 2 measure the actual mtDNA levels.

Response: We appreciate this comment, and have moved previous Figure 2 as Supplementary data of Figure 1(Fig. S2).

b) There is not clear explanation of how is the breeding scheme carried out and how often have animals been intercrossed in this way. This has to be

added to Material and Methods as is important for the interpretation of the results. Previously it was shown that extensive interbreeding causes appearance of additional phenotypes that are not directly connected with mtDNA mutation load (Ross et al., 2013).

Response: We appreciate this comment, and have added clear explanation of the breeding scheme to Material and Methods as suggested. We didn't use extensive interbreeding in our study.

As in text:

PolgA^{WT/Mut} male mice were continuously crossed to WT female mice for maintaining the population of PolgA^{WT/Mut} mice. Then, the male and female PolgA^{WT/Mut} siblings were crossed to generate three genotype mice including WT/WT*, WT/Mut** and Mut/Mut***. The same batch of male mice was used to perform experiments in accordance with relevant guidelines and regulations, which had been reviewed and approved by Guangzhou Institutes of Biomedicine and Health Ethical Committee (Approve no. 2018040).

c) The types of mtDNA mutations, deletions and insertions have to be reported. This is very important for the quality control of the data, as many mtDNA mutator mutation spectra are published over the years.

Response: We appreciate this comment, and have moved all mtDNA sequencing results as supplementary data, including mtDNA mutations in the intestine, liver and brain of POLG mutator mice (previous Fig. 2 moved into Fig. S2, and new added Fig. S3), and mtDNA mutations in the physiological aged small intestine of 20 month-old wild-type mice (new added Fig. S1).

3. MtUPR has been highly disputed as a major stress response to respiratory chain dysfunction in mammals. In *C.elegans* where mtUPR has been intensely studied HSP60 is claimed to be highly upregulated, hence used as a marker of mtUPR. In mammals, multiple studies over last 5-10 years have described mitochondrial Integrated Stress Response (mitoISR) as a mechanism by which mitochondria signal OXPHOS deficiency to the nucleus (Bao et al., 2016; Fessler et al., 2020; Guo et al., 2020; Kaspar et al., 2021; Khan et al., 2017; Kuhl et al., 2017; Mick et al., 2020; Quiros et al., 2017). This stress response is dependent on the ATF4 expression that remodels metabolism (primary toward 1C/folate metabolism) and induces the upregulation of multiple proteins, including LONP1. The authors should analyze the RNASeq data for signatures of ATF4 targets and ISR. How these data are presented in the manuscript right now, really resembles cherry picking.

Response: We appreciate this comment, and performed Kyoto Encyclopedia of Genes and Genomes (KEGG) pathway analysis of the differential genes in the intestines of Mut/Mut^{***} compared to WT/WT^{*} mice at 8 months as suggested, and revealed that mtDNA mutations trigger a retrograde mitochondria-to-nucleus communication by upregulating integrated stress response (ISR) (Fig. 6a and Supplementary Fig. 8a). The activation of ISR in Mut/Mut^{***} mice is further validated by the upregulation of eIF2 α -P and ATF5 using western blotting (Fig. 6b). *In vivo* NMN could inhibit the upregulation of eIF2 α -P and ATF5 (Fig. 6c), and the eIF2 α phosphorylation inhibitor ISRIB suppresses the upregulation of ATF5 and downregulation of β -catenin in Mut/Mut^{***} intestine (Fig. 6d,e). These results showed that NAD⁺ depletion triggers ATF5 dependent ISR activation. Using organoids system, we found ISR inhibition by ISRIR and si*Atf5* could rescue the decreased organoids formation efficiency caused by mtDNA mutations, showing the regulation of ISR upon intestinal aging (Fig. 6f, g).

As in text:

LONP1 upregulation induced by ISR activation regulates mtDNA mutation-induced aging phenotypes.

How does NAD⁺ depletion regulate Wnt/ β -catenin pathway during the intestinal aging caused by mtDNA mutations? We performed Kyoto Encyclopedia of Genes and Genomes (KEGG) pathway analysis of the up-regulated differential genes in the intestines of Mut/Mut^{***} compared to WT/WT^{*} mice at 8 months, and revealed that mtDNA mutations trigger a retrograde mitochondria-to-nucleus communication by upregulating integrated stress response (ISR) including oxidative phosphorylation, glutathione metabolism, nicotinate and nicotinamide metabolism, folate biosynthesis, PPAR signaling pathway and carbon metabolism (Fig. 6a and Supplementary Fig. 8a). Upon mitochondrial ISR, eukaryotic initiation factor 2 α subunit (eIF2 α) is phosphorylated, which subsequently results in global attenuation of cytosolic translation coincident with preferential translation of mitochondrial stress responsive transcription factors such as activating transcription factor 4 (ATF4)⁴⁴⁻⁴⁷, activating transcription factor 5 (ATF5)^{48,49} and the C/EBP Homologous Protein (CHOP)^{50,51} to restore mitochondrial function. Using western blotting, we observed upregulation of eIF2 α phosphorylation (eIF2 α -P)⁵² and ATF5 in the intestine of Mut/Mut^{***}, which could be decreased by *in vivo* NAD⁺ repletion (Fig. 6b, c and Supplementary Fig. 8b,c). Phosphorylation of eIF2 α is reported to direct ATF5 translational control in response to stress⁵³. Consistently, eIF2 α phosphorylation inhibitor ISRIB suppresses the upregulation of ATF5 in Mut/Mut^{***} intestine (Fig. 6d). These results showed that NAD⁺ depletion triggers ATF5 dependent ISR activation. ISR inhibition

increases β -catenin protein level in the intestinal crypts of Mut/Mut^{***} mice (Fig. 6e), implying its regulation on intestinal aging by restoring Wnt/ β -catenin signaling pathway. To test this, we used ISRIB and si*Atf5* to inhibit ISR in Mut/Mut^{***} intestinal crypts (Supplementary Fig. 8d), and found that both ISRIB and si*Atf5* could obviously rescue the decreased organoids formation efficiency caused by mtDNA mutations (Fig. 6f, g). Together these results indicate that NAD⁺ depletion activates ISR to inhibit Wnt/ β -catenin pathway, thus inducing the intestinal aging caused by mtDNA mutations.

4. OXPHOS deficiency in mtDNA mutator mice will cause a strong decrease in NAD⁺/NADH levels. Supplementation by NAD⁺ precursors to alleviate phenotypes of OXPHOS deficiency is not a novel concept, although it is nice to see that it works also in this system. The mechanistic connection to LONP1-HSP60 provided in this manuscript is at best indirect.

Response: We appreciate this comment, and have added more experiments to show the mechanism that links NAD⁺ depletion and LONP1/HSP60 expression (new added Fig. 6 and Fig.7) as suggested. We found that high levels of mtDNA mutations trigger an activation of ATF5 dependent ISR by NAD⁺ depletion and thus accelerate intestinal aging, providing a new mechanism linking NAD⁺ depletion and aging. We showed that the mitochondrial ISR activation upregulates UPRmt component LONP1 expression but downregulates UPRmt component HSP60, implying a specific regulation of ISR activation on the protein expression of UPRmt components. We also found that the inhibition of LONP1 overexpression by si*Lonp1*, ATF5 silence and ISRIB enhance the regenerative potential of aged intestinal crypts. Thus, we found multiple candidates for treating intestinal aging, not just preventing intestinal aging.

As in text:

NAD⁺ depletion activates ISR to regulate mtDNA mutation-induced aging phenotypes by impairing the Wnt/ β -catenin pathway.

How does NAD⁺ depletion regulate Wnt/ β -catenin pathway during the intestinal aging caused by mtDNA mutations? We performed Kyoto Encyclopedia of Genes and Genomes (KEGG) pathway analysis of the up-regulated differential genes in the intestines of Mut/Mut^{***} compared to WT/WT^{*} mice at 8 months, and revealed that mtDNA mutations trigger a retrograde mitochondria-to-nucleus communication by upregulating integrated stress response (ISR) including oxidative phosphorylation, glutathione metabolism, nicotinate and nicotinamide metabolism, folate biosynthesis, PPAR signaling pathway and carbon metabolism (Fig. 6a and Supplementary Fig. 8a). Upon mitochondrial ISR, eukaryotic initiation factor 2 α subunit (eIF2 α)

is phosphorylated, which subsequently results in global attenuation of cytosolic translation coincident with preferential translation of mitochondrial stress responsive transcription factors such as activating transcription factor 4 (ATF4)⁴⁴⁻⁴⁷, activating transcription factor 5 (ATF5)^{48,49} and The C/EBP Homologous Protein (Chop)^{50,51} to restore mitochondrial function. Using western blotting, we observed upregulation of eIF2 α phosphorylation (eIF2 α -P)⁵² and ATF5 in the intestine of Mut/Mut^{***}, which could be decreased by *in vivo* NAD⁺ repletion (Fig. 6b, c and Supplementary Fig. 8b,c). Phosphorylation of eIF2 α is reported to direct ATF5 translational control in response to stress⁵³. Consistently, eIF2 α phosphorylation inhibitor ISRIB suppresses the upregulation of ATF5 in Mut/Mut^{***} intestine (Fig. 6d). These results showed that NAD⁺ depletion triggers ATF5 dependent ISR activation. ISR inhibition increases β -catenin protein level in the intestinal crypts of Mut/Mut^{***} mice (Fig. 6e), implying its regulation on intestinal aging by restoring Wnt/ β -catenin signaling pathway. To test this, we used ISRIB and si*Atf5* to inhibit ISR in Mut/Mut^{***} intestinal crypts (Supplementary Fig. 8d), and found that both ISRIB and si*Atf5* could obviously rescue the decreased organoids formation efficiency caused by mtDNA mutations (Fig. 6f, g). Together these results indicate that NAD⁺ depletion activates ISR to inhibit Wnt/ β -catenin pathway, thus inducing the intestinal aging caused by mtDNA mutations.

LONP1 upregulation induced by ISR activation regulates mtDNA mutation-induced aging phenotypes.

As the ISR regulation factor ATF5 is also thought to be regulated by mitochondrial import efficiency to activate mammalian unfolded protein response (UPRmt)⁵⁴, we assayed the protein expression of UPRmt components including Lon protease (LONP1), caseinolytic peptidase P (ClpP) and Heat shock protein 60 (HSP60) in Mut/Mut^{***} intestine with an ATF5-dependent ISR activation. Interestingly, we observed that only LONP1, a key mitochondrial protease for clearing unfolded proteins upon mitochondrial stress response⁵⁵, is obviously increased while mitochondrial amount is not altered in Mut/Mut^{***} intestinal crypts (Fig. 7a and Supplementary Fig. 9a). Both ISRIB and si*Atf5* could inhibit the upregulation of LONP1 in Mut/Mut^{***} intestinal crypts (Fig. 7b, c), demonstrating a regulation of ATF5 dependent ISR on LONP1 expression in the intestinal aging caused by mtDNA mutations. To further test the role of LONP1 in the intestinal aging, we used a validated small interfering RNA for *Lonp1* (Supplementary Fig. 9b), and found that the inhibition of LONP1 overexpression by *Lonp1* silencing could promote colony formation efficiency in Mut/Mut^{***} intestinal crypts (Fig. 7d and Supplementary Fig. 9c), showing an important role of LONP1 overexpression in regulating the intestinal aging. In addition, the downregulation of HSP60 (Fig. 7a and Supplementary Fig. 9d) could be restored by the inhibition of eIF2 α phosphorylation, indicating a regulation of ISR activation on the protein expression of UPRmt components (Supplementary Fig. 9e). Together, these

data indicate LONP1 upregulation induced by ISR activation regulates the aging intestinal phenotypes caused by mtDNA mutations.

5. Results on the figure 7 are really baffling. First of all, HSP60 is by far the most abundant protein in mitochondria. Therefore, it is not clear, why it suddenly disappears from 12M old samples (immunofluorescence). Are mitochondria in all genotypes suddenly gone? The same goes for the immunohistochemistry upon NMN treatment. What about naïve WT samples of the same ages? The authors should at least provide western blot of HSP60 and LONP1 at all ages and all genotypes. Including naïve WT control. Actually, it would be very helpful to compare proteome of all different samples and ages for the observed changes. Additional western blots used for the quantification should be provided.

Response: We appreciate this comment, have revised the presentation about HSP60 FI, have added experiments to check mitochondrial amount in naïve WT, WT/WT*, WT/Mut** and Mut/Mut*** mice (new added Fig. S9a), and have added experiments to show the mechanism that regulates LONP1 and HSP60 expression (new added Fig. 7) as suggested.

To avoid the overexposure of HSP60 FI in all different samples and ages, the exposure of HSP60 FI in WT/WT* mice at 8 months, which is highest, was used for all samples to perform HSP60 imaging. Thus, HSP60 FI of 12 month Mut/Mut*** seems weak but still have more than 20% FI of control, which is removed for the possible puzzle. We further checked mitochondrial amount of small intestine in naïve WT control, WT/WT*, WT/Mut** and Mut/Mut*** using anti-TOM20 western blotting (Fig. S9a), and found that mitochondrial amount is not changed by mtDNA mutations at 8 months. In addition, the anti-TOM20 IF also showed that mitochondria is not gone in the small intestine of Mut/Mut*** mice at 12 months. We further showed that the mitochondrial ISR activation upregulates UPRmt component LONP1 expression but downregulates UPRmt component HSP60, implying a specific regulation of ISR activation on the protein expression of UPRmt components.

As in text:

As the ISR regulation factor ATF5 is also thought to be regulated by mitochondrial import efficiency to activate mammalian unfolded protein response (UPRmt)⁵⁴, we assayed the protein expression of UPRmt components including Lon protease (LONP1), caseinolytic peptidase P (ClpP) and Heat shock protein 60 (HSP60) in Mut/Mut*** intestine with an ATF5-dependent ISR activation. Interestingly, we observed that only LONP1, a key mitochondrial protease for clearing unfolded proteins upon mitochondrial stress response⁵⁵, is obviously increased while mitochondrial amount is not

altered in Mut/Mut^{***} intestinal crypts (Fig. 7a and Supplementary Fig. 9a). Both ISRIB and si*Atf5* could inhibit the upregulation of LONP1 in Mut/Mut^{***} intestinal crypts (Fig. 7b, c), demonstrating a regulation of ATF5 dependent ISR on LONP1 expression in the intestinal aging caused by mtDNA mutations. To further test the role of LONP1 in the intestinal aging, we used a validated small interfering RNA for *Lonp1* (Supplementary Fig. 9b), and found that the inhibition of LONP1 overexpression by *Lonp1* silencing could promote colony formation efficiency in Mut/Mut^{***} intestinal crypts (Fig. 7d and Supplementary Fig. 9c), showing an important role of LONP1 overexpression in regulating the intestinal aging. In addition, the downregulation of HSP60 (Fig. 7a and Supplementary Fig. 9d) could be restored by the inhibition of eIF2 α phosphorylation, indicating a regulation of ISR activation on the protein expression of UPR^{mt} components (Supplementary Fig. 9e). Together, these data indicate LONP1 upregulation induced by ISR activation regulates the aging intestinal phenotypes caused by mtDNA mutations.

6. The very last result panel of figure 7 (7D) is even more puzzling. The depletion of LONP1 provides rescue for the organoids in culture! Mutations in LONP1 cause several devastating diseases in patients and the loss of LONP1 in mice causes very early embryonic lethality. LONP1 has been shown to be THE major mitochondrial protease important for the removal of unfolded and misfolded proteins. Therefore, it is not clear why would its removal improve phenotypes in the mtDNA mutator model.

Response: We appreciate this comment, have added experiments to show the LONP1 expression in the aged Mut/Mut^{***} intestine treated with si*Lonp1* (Fig. S9c), and revised the presentation more clearly as suggested. We observed the up-regulation of LONP1 in the intestine of Mut/Mut^{***} mice, which is appropriate 3 times that of WT/WT^{*} mice. Thus, we used the si*Lonp1* to inhibit the overexpression of LONP1 in the intestinal crypts of Mut/Mut^{***} mice. We found that LONP1 expression of Mut/Mut^{***} mice using si*Lonp1* decreases to 70% of control (Fig. S9c). This indicates a restoration of LONP1 expression but not a complete LONP1 inhibition. The inhibition of LONP1 overexpression improve aging phenotypes in the mtDNA mutator model. Thus, LONP1 upregulation induced by ISR activation regulates mtDNA mutation-induced aging phenotypes. LONP1 has been described to degrade cytochrome c oxidase subunits IV and V in the complex IV of oxidative phosphorylation 67-69. The upregulation of LONP1 provides a possible explanation for the observed mitochondrial dysfunction caused by mtDNA mutations, which further exacerbates the intestinal aging phenotypes.

As in text:

As the ISR regulation factor ATF5 is also thought to be regulated by mitochondrial import efficiency to activate mammalian unfolded protein response (UPR_{mt})⁵⁴, we assayed the protein expression of UPR_{mt} components including Lon protease (LONP1), caseinolytic peptidase P (ClpP) and Heat shock protein 60 (HSP60) in Mut/Mut^{***} intestine with an ATF5-dependent ISR activation. Interestingly, we observed that only LONP1, a key mitochondrial protease for clearing unfolded proteins upon mitochondrial stress response⁵⁵, is obviously increased while mitochondrial amount is not altered in Mut/Mut^{***} intestinal crypts (Fig. 7a and Supplementary Fig. 9a). Both ISRIB and si*Atf5* could inhibit the upregulation of LONP1 in Mut/Mut^{***} intestinal crypts (Fig. 7b, c), demonstrating a regulation of ATF5 dependent ISR on LONP1 expression in the intestinal aging caused by mtDNA mutations. To further test the role of LONP1 in the intestinal aging, we used a validated small interfering RNA for *Lonp1* (Supplementary Fig. 9b), and found that the inhibition of LONP1 overexpression by *Lonp1* silencing could promote colony formation efficiency in Mut/Mut^{***} intestinal crypts (Fig. 7d and Supplementary Fig. 9c), showing an important role of LONP1 overexpression in regulating the intestinal aging. In addition, the downregulation of HSP60 (Fig. 7a and Supplementary Fig. 9d) could be restored by the inhibition of eIF2 α phosphorylation, indicating a regulation of ISR activation on the protein expression of UPR_{mt} components (Supplementary Fig. 9e). Together, these data indicate LONP1 upregulation induced by ISR activation regulates the aging intestinal phenotypes caused by mtDNA mutations.

7. The manuscript does not address the function of mitochondria in the investigated models. At least some data on the level and function of OXPHOS should be provided.

Response: We appreciate this comment, and have detected mitochondrial inner membrane potential using JC-1 staining and ROS level using DCF staining in the intestinal crypts of WT/WT^{*}, WT/Mut^{**} and Mut/Mut^{***} mice (new added Fig. S4c, d) as suggested. As expected, the intestinal crypt cells in Mut/Mut^{***} mice exhibit a lower mitochondrial inner membrane potential and higher ROS level than that in WT/WT^{*} mice, indicating that high levels of mtDNA mutations result in mitochondrial dysfunction during the intestinal aging (Supplementary Fig. 4c, d).

As in text:

Most enriched GO terms are involved in mitochondrial respiratory function and metabolic process, demonstrating an alteration of mitochondrial function caused by mtDNA mutations. As expected, the intestinal crypt cells in Mut/Mut^{***} mice exhibit a lower mitochondrial inner membrane potential and higher ROS level than that in WT/WT^{*} mice, indicating that high levels of

mtDNA mutations result in mitochondrial dysfunction during the intestinal aging (Supplementary Fig. 4c, d).

- Bao, X.R., Ong, S.E., Goldberger, O., Peng, J., Sharma, R., Thompson, D.A., Vafai, S.B., Cox, A.G., Marutani, E., Ichinose, F., et al. (2016). Mitochondrial dysfunction remodels one-carbon metabolism in human cells. *Elife* 5. 10.7554/eLife.10575.
- Dayama, G., Emery, S.B., Kidd, J.M., and Mills, R.E. (2014). The genomic landscape of polymorphic human nuclear mitochondrial insertions. *Nucleic Acids Res* 42, 12640-12649. 10.1093/nar/gku1038. **25348406**
- Fessler, E., Eckl, E.M., Schmitt, S., Mancilla, I.A., Meyer-Bender, M.F., Hanf, M., Philippou-Massier, J., Krebs, S., Zischka, H., and Jae, L.T. (2020). A pathway coordinated by DELE1 relays mitochondrial stress to the cytosol. *Nature* 579, 433-437. 10.1038/s41586-020-2076-4. 32132706
- Guo, X., Aviles, G., Liu, Y., Tian, R., Unger, B.A., Lin, Y.T., Wiita, A.P., Xu, K., Correia, M.A., and Kampmann, M. (2020). Mitochondrial stress is relayed to the cytosol by an OMA1-DELE1-HRI pathway. *Nature* 579, 427-432. 10.1038/s41586-020-2078-2.
- Isokallio, M.A., and Stewart, J.B. (2021). High-Throughput Detection of mtDNA Mutations Leading to tRNA Processing Errors. *Methods Mol Biol* 2192, 117-132. 10.1007/978-1-0716-0834-0_10.
- Kaspar, S., Oertlin, C., Szczepanowska, K., Kukat, A., Senft, K., Lucas, C., Brodesser, S., Hatzoglou, M., Larsson, O., Topisirovic, I., and Trifunovic, A. (2021). Adaptation to mitochondrial stress requires CHOP-directed tuning of ISR. *Sci Adv* 7. 10.1126/sciadv.abf0971.
- Khan, N.A., Nikkanen, J., Yatsuga, S., Jackson, C., Wang, L., Pradhan, S., Kivela, R., Pessia, A., Velagapudi, V., and Suomalainen, A. (2017). mTORC1 Regulates Mitochondrial Integrated Stress Response and Mitochondrial Myopathy Progression. *Cell Metab* 26, 419-428 e415. 10.1016/j.cmet.2017.07.007.
- Kuhl, I., Miranda, M., Atanassov, I., Kuznetsova, I., Hinze, Y., Mourier, A., Filipovska, A., and Larsson, N.G. (2017). Transcriptomic and proteomic landscape of mitochondrial dysfunction reveals secondary coenzyme Q deficiency in mammals. *Elife* 6. 10.7554/eLife.30952.
- Mick, E., Titov, D.V., Skinner, O.S., Sharma, R., Jourdain, A.A., and Mootha, V.K. (2020). Distinct mitochondrial defects trigger the integrated stress response depending on the metabolic state of the cell. *Elife* 9. 10.7554/eLife.49178.
- Quiros, P.M., Prado, M.A., Zamboni, N., D'Amico, D., Williams, R.W., Finley, D., Gygi, S.P., and Auwerx, J. (2017). Multi-omics analysis identifies ATF4 as a key regulator of the mitochondrial stress response in mammals. *J Cell Biol* 216, 2027-2045. 10.1083/jcb.201702058.
- Ross, J.M., Stewart, J.B., Hagstrom, E., Brene, S., Mourier, A., Coppotelli, G., Freyer, C., Lagouge, M., Hoffer, B.J., Olson, L., and Larsson, N.G. (2013).

Germline mitochondrial DNA mutations aggravate ageing and can impair brain development. *Nature* 501, 412-415. 10.1038/nature12474.

Reviewer #2 (Remarks to the Author):

Yang et al show that mitochondrial mutator mice have increased villus length and reduced basal crypt number, reduced ISC number, defective b-catenin activation, reduced NAD. NAD repletion partially rescues the defects. The mutant mice failed to upregulate mitophagy at mid age independent of NAD. HSP60 increases with aging, consistent with increased mitochondrial protein folding stress with aging. However, HSP60 is reduced in mutant mice and can be rescued by NAD repletion. Mutant mice have increased LONP1 instead. LONP1 knockdown increases HSP60 and rescues the intestine defects in vitro.

Fig 1b. There is little difference in villus length and basal crypt number in WT mice of various ages. It is unclear whether these are indeed aging related changes based on these data, raising the question of the physiological significance of the changes in mitochondrial mutator mice.

Response: We appreciate this comment, and have added experiments (Fig.S1a-d) to detect the aging biomarker and mtDNA mutations in small intestine of young (3 months) and old mice (20 months) as suggested. As well as the small intestine of POLG mutator mice, the small intestine of aged wild-type mice also showed higher expression of CDKN1A/p21 (a well-known senescence marker) and shorter telomere length (Supplementary Fig.1a, b), validating the intestinal aging in both POLG mutator mice and 20-month-old wild-type mice. The aged intestine of wild-type mice accumulates more mtDNA mutations, primarily low-frequency (less than 0.05) point mutations (Supplementary Fig.1c, d), demonstrating a link between mtDNA mutations and the physiological intestinal aging. Notably, the aged intestine of POLG mutator mice also show a similar but faster accumulation of low-frequency mtDNA point mutations than the aged intestine of wild-type mice, providing a suitable model for investigating the roles of mtDNA mutations in the intestinal aging.

As in text:

As well as aged human clinical intestinal samples²¹⁻²³, small intestine of aged mouse intestine, marked by higher expression of CDKN1A/p21 (a well-known senescence marker) and shorter telomere length (Supplementary Fig.1a, b), also accumulates more mtDNA mutations, primarily low-frequency (less than

0.05) point mutations (Supplementary Fig.1c, d), demonstrating a link between mtDNA mutations and the physiological intestinal aging.

As compared to WT/WT mice, at 8 months Mut/Mut*** mice accumulate much more of point mutations, primarily low-frequency (less than 0.05) point mutations which is also observed in the small intestine of aged wild-type mice (Supplementary Fig. 2a-d).

Fig 2a. There is no difference in point mutation, insertion and deletion in WT mice of various ages. These data do not support increased mtDNA mutation with age, again raising the question of the physiological significance of the changes in mitochondrial mutator mice.

Response: We appreciate this comment, and have added experiments (Fig.S1a-d) to detect the mtDNA mutations in small intestinal of young (3 months) and old mice (20 months). As well as aged human clinical intestinal samples, small intestine of aged mouse intestine, marked by higher expression of CDKN1A/p21 (a well-known senescence marker) and shorter telomere length (Supplementary Fig.1a, b), also accumulates more mtDNA mutations, primarily low-frequency (less than 0.05) point mutations (Supplementary Fig.1c, d), demonstrating a link between mtDNA mutations and the physiological intestinal aging. Notably, the aged intestine of POLG mutator mice show a similar but faster accumulation of low-frequency mtDNA point mutations than the aged intestine of wild-type mice, providing a suitable model for investigating the roles of mtDNA mutations in the intestinal aging.

As in text:

As well as aged human clinical intestinal samples²¹⁻²³, small intestine of aged mouse intestine, marked by higher expression of CDKN1A/p21 (a well-known senescence marker) and shorter telomere length (Supplementary Fig.1a, b), also accumulates more mtDNA mutations, primarily low-frequency (less than 0.05) point mutations (Supplementary Fig.1c, d), demonstrating a link between mtDNA mutations and the physiological intestinal aging.

As compared to WT/WT mice, at 8 months Mut/Mut*** mice accumulate much more of point mutations, primarily low-frequency (less than 0.05) point mutations which is also observed in the small intestine of aged wild-type mice (Supplementary Fig. 2a-d).

Fig 7e. The authors propose that low mtDNA mutations result in mitophagy. There is no evidence to support this. Fig 5a show increased mitophagy in WT mice at 8 months old. But these mice do not have increased mtDNA mutation (Fig 2) compared WT mice at 3 months old. The authors propose mitophagy

occurs upstream of NAD depletion and prevents mtDNA mutation and aging. But there is no evidence to support it. To support this claim, the authors need to active mitophagy to rescue these defects and inactivate mitophagy to speed up these defects.

Response: We appreciate this comment, have moved previous Fig. 5 about mitophagy as supplementary data (Fig. S6), and have revised the overstated presentation about mitophagy as suggested. NAD⁺ repletion *in vivo* doesn't rescue mitophagy deficiency or mtDNA mutations in the intestinal crypts of Mut/Mut^{***} mice at 8 months (Supplementary Fig. 6b, c), indicating that the intestinal aging induced by NAD⁺ depletion is independent of mitophagy. This indicated the other regulation mechanism that functions in the intestinal aging by NAD⁺ depletion, and we further showed the ISR activation that links NAD⁺ depletion and intestinal aging.

As in text:

As NAD⁺ depletion has been reported to compromise mitophagy^{41,42}, it raises a possibility that NAD⁺ repletion could activate mitophagy to remove mutated mtDNAs and thus alleviates the aging phenotypes of small intestine caused by mtDNA mutations. By detecting mitophagy using anti-LC3B and anti-TOM20 IF in the intestine crypts of WT/WT^{*}, WT/Mut^{**} and Mut/Mut^{***} mice at 3, 8 and 12 months, we indeed observed compromised mitophagy in Mut/Mut^{***} mice at 8 months (Supplementary Fig. 6a). However, NAD⁺ repletion *in vivo* doesn't rescue mitophagy deficiency or mtDNA mutations in the intestinal crypts of Mut/Mut^{***} mice at 8 months (Supplementary Fig. 6b, c), indicating that the intestinal aging induced by NAD⁺ depletion is independent of mitophagy.

Another major issue is the mechanism. The authors propose high levels of mitochondrial mutation results in NAD depletion, which causes increased LONP1, and in turn causes decreased HSP60. First, how does NAD depletion cause increased LONP1? Is LONP1 expression dependent on NAD? Second, there seems to be some confusion in the concept of mitochondrial unfolded protein response. Mitochondrial unfolded protein response includes the transcriptional induction of mitochondrial chaperones such as HSP60 and proteases such as LONP1 and ClpP.

Response: We appreciate this comment, and have added experiments to reveal the mechanism that links NAD⁺ depletion and LONP1-HSP60 (new added Fig. 6 and Fig. 7) as suggested.

We showed in new added Figure 6 that NAD⁺ depletion activates ATF5 dependent ISR to inhibit Wnt/β-catenin pathway, thus inducing the intestinal aging caused by mtDNA mutations. Upon mitochondrial ISR, eukaryotic initiation factor 2 α subunit (eIF2α) is phosphorylated, which subsequently results in global attenuation of cytosolic translation coincident with preferential translation of mitochondrial stress responsive transcription factors such as activating transcription factor 4 (ATF4), activating transcription factor 5 (ATF5) and the C/EBP Homologous Protein (CHOP) to restore mitochondrial function. In the new added Fig. 7, we found that the mitochondrial ISR activation upregulates UPRmt component LONP1 expression by ATF5 but downregulates UPRmt component HSP60 by eIF2α phosphorylation. This implies a specific regulation of ISR activation on the protein expression of UPRmt components.

As in text:

NAD⁺ depletion activates ISR to regulate mtDNA mutation-induced aging phenotypes by impairing the Wnt/β-catenin pathway.

How does NAD⁺ depletion regulate Wnt/β-catenin pathway during the intestinal aging caused by mtDNA mutations? We performed Kyoto Encyclopedia of Genes and Genomes (KEGG) pathway analysis of the up-regulated differential genes in the intestines of Mut/Mut^{***} compared to WT/WT^{*} mice at 8 months, and revealed that mtDNA mutations trigger a retrograde mitochondria-to-nucleus communication by upregulating integrated stress response (ISR) including oxidative phosphorylation, glutathione metabolism, nicotinate and nicotinamide metabolism, folate biosynthesis, PPAR signaling pathway and carbon metabolism (Fig. 6a and Supplementary Fig. 8a). Upon mitochondrial ISR, eukaryotic initiation factor 2 α subunit (eIF2α) is phosphorylated, which subsequently results in global attenuation of cytosolic translation coincident with preferential translation of mitochondrial stress responsive transcription factors such as activating transcription factor 4 (ATF4)⁴⁴⁻⁴⁷, activating transcription factor 5 (ATF5)^{48,49} and the C/EBP Homologous Protein (CHOP)^{50,51} to restore mitochondrial function. Using western blotting, we observed upregulation of eIF2α phosphorylation (eIF2α-P)⁵² and ATF5 in the intestine of Mut/Mut^{***}, which could be decreased by *in vivo* NAD⁺ repletion (Fig. 6b, c and Supplementary Fig. 8b,c). Phosphorylation of eIF2α is reported to direct ATF5 translational control in response to stress⁵³. Consistently, eIF2α phosphorylation inhibitor ISRIB suppresses the upregulation of ATF5 in Mut/Mut^{***} intestine (Fig. 6d). These results showed that NAD⁺ depletion triggers ATF5 dependent ISR activation. ISR inhibition increases β-catenin protein level in the intestinal crypts of Mut/Mut^{***} mice (Fig. 6e), implying its regulation on intestinal aging by restoring Wnt/β-catenin signaling pathway. To test this, we used ISRIB and si*Atf5* to inhibit ISR in Mut/Mut^{***} intestinal crypts (Supplementary Fig. 8d), and found that both

ISRIB and si*Atf5* could obviously rescue the decreased organoids formation efficiency caused by mtDNA mutations (Fig. 6f, g). Together these results indicate that NAD⁺ depletion activates ATF5 dependent ISR to inhibit Wnt/ β -catenin pathway, thus inducing the intestinal aging caused by mtDNA mutations.

LONP1 upregulation induced by ISR activation regulates mtDNA mutation-induced aging phenotypes.

As the ISR regulation factor ATF5 is also thought to be regulated by mitochondrial import efficiency to activate mammalian unfolded protein response (UPRmt)⁵⁴, we assayed the protein expression of UPRmt components including Lon protease (LONP1), caseinolytic peptidase P (ClpP) and Heat shock protein 60 (HSP60) in Mut/Mut^{***} intestine with an ATF5-dependent ISR activation. Interestingly, we observed that only LONP1, a key mitochondrial protease for clearing unfolded proteins upon mitochondrial stress response⁵⁵, is obviously increased while mitochondrial amount is not altered in Mut/Mut^{***} intestinal crypts (Fig. 7a and Supplementary Fig. 9a). Both ISRIB and si*Atf5* could inhibit the upregulation of LONP1 in Mut/Mut^{***} intestinal crypts (Fig. 7b, c), demonstrating a regulation of ATF5 dependent ISR on LONP1 expression in the intestinal aging caused by mtDNA mutations. To further test the role of LONP1 in the intestinal aging, we used a validated small interfering RNA for *Lonp1* (Supplementary Fig. 9b), and found that the inhibition of LONP1 overexpression by *Lonp1* silencing could promote colony formation efficiency in Mut/Mut^{***} intestinal crypts (Fig. 7d and Supplementary Fig. 9c), showing an important role of LONP1 overexpression in regulating the intestinal aging. In addition, the downregulation of HSP60 (Fig. 7a and Supplementary Fig. 9d) could be restored by the inhibition of eIF2 α phosphorylation, indicating a regulation of ISR activation on the protein expression of UPRmt components (Supplementary Fig. 9e). Together, these data indicate LONP1 upregulation induced by ISR activation regulates the aging intestinal phenotypes caused by mtDNA mutations.

Fig 7d. It is hard to imaging without the protection of LONP1, it will be beneficial to the cells or the tissues. Has the authors done the rescue experiment *in vivo*?

Response: We appreciate this comment, have added experiments to showed the LONP1 expression in the aged intestinal of Mut/Mut^{***} mice treated with si*Lonp1* (Fig. S9c), and have revised the presentation more clearly as suggested. We observed the up-regulation of LONP1 in the intestine of Mut/Mut^{***} mice, which is appropriate 3 times that of WT/WT* mice. Thus, we used the si*Lonp1* to inhibit the overexpression of LONP1 in the intestinal crypts of Mut/Mut^{***} mice. We found that LONP1 expression of Mut/Mut^{***} mice

using *siLonp1* decreases to 70% of control (Fig. S9c), and restoring normal LONP1 expression by inhibiting LONP1 overexpression improve aging phenotypes in the mtDNA mutator model. We showed LONP1 upregulation induced by ISR activation regulates mtDNA mutation-induced aging phenotypes. LONP1 has been described to degrade cytochrome c oxidase subunits IV and V in the complex IV of oxidative phosphorylation. The upregulation of LONP1 provides a possible explanation for the observed mitochondrial dysfunction caused by mtDNA mutations, which further exacerbates the intestinal aging phenotypes.

As in text:

As the ISR regulation factor ATF5 is also thought to be regulated by mitochondrial import efficiency to activate mammalian unfolded protein response (UPR_{mt})⁵⁴, we assayed the protein expression of UPR_{mt} components including Lon protease (LONP1), caseinolytic peptidase P (ClpP) and Heat shock protein 60 (HSP60) in Mut/Mut^{***} intestine with an ATF5-dependent ISR activation. Interestingly, we observed that only LONP1, a key mitochondrial protease for clearing unfolded proteins upon mitochondrial stress response⁵⁵, is obviously increased while mitochondrial amount is not altered in Mut/Mut^{***} intestinal crypts (Fig. 7a and Supplementary Fig. 9a). Both ISRIB and *siAtf5* could inhibit the upregulation of LONP1 in Mut/Mut^{***} intestinal crypts (Fig. 7b, c), demonstrating a regulation of ATF5 dependent ISR on LONP1 expression in the intestinal aging caused by mtDNA mutations. To further test the role of LONP1 in the intestinal aging, we used a validated small interfering RNA for *Lonp1* (Supplementary Fig. 9b), and found that the inhibition of LONP1 overexpression by *Lonp1* silencing could promote colony formation efficiency in Mut/Mut^{***} intestinal crypts (Fig. 7d and Supplementary Fig. 9c), showing an important role of LONP1 overexpression in regulating the intestinal aging. In addition, the downregulation of HSP60 (Fig. 7a and Supplementary Fig. 9d) could be restored by the inhibition of eIF2 α phosphorylation, indicating a regulation of ISR activation on the protein expression of UPR_{mt} components (Supplementary Fig. 9e). Together, these data indicate LONP1 upregulation induced by ISR activation regulates the aging intestinal phenotypes caused by mtDNA mutations.

Reviewer #3 (Remarks to the Author):

Mitochondrial dysfunction plays an important role in cell senescence and organism aging. In the manuscript, Yang et al identified an AD⁺-LONP1-UPR_{mt} pathway that underlies intestinal aging caused by mitochondrial DNA mutations. Using polymerase gamma (POLG) mutator

mice, the authors constructed an intestinal aging model and tried to elucidate the underlying mechanisms by which mtDNA mutations causes aging of LGR5-expressing intestinal stem cells. They found that 1) low rates of mtDNA mutation activate mitophagy to curb the mutational load; 2) high levels of mtDNA mutations result in NAD⁺ depletion and altered LONP1-dependent UPR_{mt} pathway; 3) the intestinal aging phenotype can be reversed by supplementation with NAD⁺ precursor, NMN; 4) ISC aging is accompanied by a decrease in Wnt-beta-catenin activation; 5) Lonp1 silencing can reverse the aging phenotypes. Overall, the manuscript provides some valuable information regarding ISC aging, however, it is premature to publish these results in Nat Commun at this stage.

Major concerns:

1. In the aging model, at the late stages, the authors show that ISCs are almost gone and the enterocytes undergo extensive apoptosis. Intriguingly, the authors also show there is an increase in the size of the villi. How could this happen?

Response: We appreciate this comment, and have added anti-Notch1 IF experiments to detect the Notch signaling that could explain for the increased villi size in Mut/Mut^{***} mice at 8 months (new added Fig. S7c) as suggested. The activation of Notch signaling could induce Paneth cells to proliferate and differentiate into villus epithelial cells. We did find the activation of Notch signaling in Mut/Mut^{***} mice at 8 months, providing an explanation for the increase of villus length in the aged intestine lacking LGR5-expressing ISCs and Wnt/ β -catenin pathway.

As in text:

As the activation of Notch signaling could induce Paneth cells to proliferate and differentiate into villus epithelial cells ⁴³, we performed anti-Notch IF and did observe the activation of Notch signal in the intestinal crypt base of Mut/Mut^{***} mice at 8 months (Supplementary Fig. 7c), providing a possible explanation for the increase of villus length in the aged intestine lacking LGR5-expressing ISCs and Wnt/ β -catenin pathway.

2. The authors show a decrease in beta-catenin activation in the aging model, which is likely the most important driving force for ISC renewal. Wnt molecules are believed to be supplied by the niche of ISCs: the Gli1⁺ or Foxl1⁺ stromal cells or Paneth cells. Do mitochondrial DNA mutations affect the niche cells or the secretion of Wnt molecules by the niche cells?

Response: We appreciate this comment, have added anti-Wnt4 western blotting experiments to show the secretion of Wnt molecules in mtDNA mutator

mice (Fig. 5d, e), and have added anti-Foxl1 and anti-Gli1 IF experiments to detect the niche cells of ISCs in mtDNA mutator mice (Supplementary Fig. 7a-b) as suggested. We found Wnt4 is decreased in Mut/Mut^{***} intestinal crypts, which could be restored by NAD⁺ repletion (Fig. 5d, e). Furthermore, we observed the downregulation of Foxl1 but not Gli1 in Mut/Mut^{***} intestine, implying a possible reduction of Foxl1⁺ cells caused by mtDNA mutations (Supplementary Fig. 7a-b).

As in text:

For the requirement of Wnt molecules secreted from the niche cells including Gli1⁺, Foxl1⁺ stromal and Paneth cells as self-renewal factors to support ISCs³⁻⁵, we assayed the protein expression of Wnt4, one well-known intestinal Wnt molecule 4, in the intestinal crypts of WT/WT^{*}, WT/Mut^{**} and Mut/Mut^{***} mice at 8 months. We found it is decreased in Mut/Mut^{***} intestinal crypts, which could be restored by NAD⁺ repletion (Fig. 5d, e). Furthermore, we observed the downregulation of Foxl1 but not Gli1 in Mut/Mut^{***} intestine, implying a possible reduction of Foxl1⁺ cells caused by mtDNA mutations (Supplementary Fig. 7a-b). Taken together, NAD⁺ depletion caused by high levels of mtDNA mutations induces the decline of LGR5-expressing ISCs via impairing the Wnt/ β -catenin pathway.

3. The link between NAD depletion and the Wnt-Catenin pathway and the link between NAD depletion and UPR need to be established. These findings appear to be fragmented at this stage.

Response: We appreciate this comment, and have added experiments (new added Fig. 6 and Fig. 7) to show the ATF5-dependent ISR activation that links NAD⁺ depletion and Wnt-Catenin pathway, and links NAD⁺ depletion and the protein expression of UPRmt components. In brief, we found that NAD⁺ depletion activates ISR to regulate mtDNA mutation-induced aging phenotypes by impairing Wnt/ β -catenin pathway. We found that the mitochondrial ISR activation upregulates UPRmt component LONP1 expression but downregulates UPRmt component HSP60. LONP1 upregulation induced by ISR activation regulates mtDNA mutation-induced aging phenotypes.

NAD⁺ depletion activates ISR to regulate mtDNA mutation-induced aging phenotypes by impairing the Wnt/ β -catenin pathway.

How does NAD⁺ depletion regulate Wnt/ β -catenin pathway during the intestinal aging caused by mtDNA mutations? We performed Kyoto Encyclopedia of Genes and Genomes (KEGG) pathway analysis of the

up-regulated differential genes in the intestines of Mut/Mut^{***} compared to WT/WT^{*} mice at 8 months, and revealed that mtDNA mutations trigger a retrograde mitochondria-to-nucleus communication by upregulating integrated stress response (ISR) including oxidative phosphorylation, glutathione metabolism, nicotinate and nicotinamide metabolism, folate biosynthesis, PPAR signaling pathway and carbon metabolism (Fig. 6a and Supplementary Fig. 8a). Upon mitochondrial ISR, eukaryotic initiation factor 2 α subunit (eIF2 α) is phosphorylated, which subsequently results in global attenuation of cytosolic translation coincident with preferential translation of mitochondrial stress responsive transcription factors such as activating transcription factor 4 (ATF4)⁴⁴⁻⁴⁷, activating transcription factor 5 (ATF5)^{48,49} and the C/EBP Homologous Protein (CHOP)^{50,51} to restore mitochondrial function. Using western blotting, we observed upregulation of eIF2 α phosphorylation (eIF2 α -P)⁵² and ATF5 in the intestine of Mut/Mut^{***}, which could be decreased by *in vivo* NAD⁺ repletion (Fig. 6b, c and Supplementary Fig. 8b,c). Phosphorylation of eIF2 α is reported to direct ATF5 translational control in response to stress⁵³. Consistently, eIF2 α phosphorylation inhibitor ISRIB suppresses the upregulation of ATF5 in Mut/Mut^{***} intestine (Fig. 6d). These results showed that NAD⁺ depletion triggers ATF5 dependent ISR activation. ISR inhibition increases β -catenin protein level in the intestinal crypts of Mut/Mut^{***} mice (Fig. 6e), implying its regulation on intestinal aging by restoring Wnt/ β -catenin signaling pathway. To test this, we used ISRIB and si*Atf5* to inhibit ISR in Mut/Mut^{***} intestinal crypts (Supplementary Fig. 8d), and found that both ISRIB and si*Atf5* could obviously rescue the decreased organoids formation efficiency caused by mtDNA mutations (Fig. 6f, g). Together these results indicate that NAD⁺ depletion activates ATF5 dependent ISR to inhibit Wnt/ β -catenin pathway, thus inducing the intestinal aging caused by mtDNA mutations.

LONP1 upregulation induced by ISR activation regulates mtDNA mutation-induced aging phenotypes.

As the ISR regulation factor ATF5 is also thought to be regulated by mitochondrial import efficiency to activate mammalian unfolded protein response (UPRmt)⁵⁴, we assayed the protein expression of UPRmt components including Lon protease (LONP1), caseinolytic peptidase P (ClpP) and Heat shock protein 60 (HSP60) in Mut/Mut^{***} intestine with an ATF5-dependent ISR activation. Interestingly, we observed that only LONP1, a key mitochondrial protease for clearing unfolded proteins upon mitochondrial stress response⁵⁵, is obviously increased while mitochondrial amount is not altered in Mut/Mut^{***} intestinal crypts (Fig. 7a and Supplementary Fig. 9a). Both ISRIB and si*Atf5* could inhibit the upregulation of LONP1 in Mut/Mut^{***} intestinal crypts (Fig. 7b, c), demonstrating a regulation of ATF5 dependent ISR on LONP1 expression in the intestinal aging caused by mtDNA mutations. To further test the role of LONP1 in the intestinal aging, we used a validated

small interfering RNA for *Lonp1* (Supplementary Fig. 9b), and found that the inhibition of LONP1 overexpression by *Lonp1* silencing could promote colony formation efficiency in Mut/Mut^{***} intestinal crypts (Fig. 7d and Supplementary Fig. 9c), showing an important role of LONP1 overexpression in regulating the intestinal aging. In addition, the downregulation of HSP60 (Fig. 7a and Supplementary Fig. 9d) could be restored by the inhibition of eIF2 α phosphorylation, indicating a regulation of ISR activation on the protein expression of UPR_{mt} components (Supplementary Fig. 9e). Together, these data indicate LONP1 upregulation induced by ISR activation regulates the aging intestinal phenotypes caused by mtDNA mutations.

4. It will be nice if the authors can provide some genetic data to verify the pathway underlying mtDNA mutations-induced ISC aging.

Response: We appreciate this comment, and have added genetic data to verify the Wnt/ β -catenin signaling (New added Fig. 4a) and ISR pathway (Fig. 6a and Fig. S8a) underlying mtDNA mutations-induced ISC aging as suggested.

As in text:

Due to the important roles of signaling pathways including Wnt, Notch, Notum, PI3K/Akt and TGF- β pathways in regulating the development process of intestinal epithelial cells, we assayed these signaling pathways and their correlation with NAD⁺ depletion during the intestinal aging. Using RNA-Seq analysis, we found that only differential genes associated with Wnt/ β -catenin signaling are enriched in Mut/Mut^{***} intestine comparing to WT/WT* at 8 months, which are down-regulated by mtDNA mutations (Fig. 4a).

How does NAD⁺ depletion regulate Wnt/ β -catenin pathway during the intestinal aging caused by mtDNA mutations? We performed Kyoto Encyclopedia of Genes and Genomes (KEGG) pathway analysis of the up-regulated differential genes in the intestines of Mut/Mut^{***} compared to WT/WT* mice at 8 months, and revealed that mtDNA mutations trigger a retrograde mitochondria-to-nucleus communication by upregulating integrated stress response (ISR) including oxidative phosphorylation, glutathione metabolism, nicotinate and nicotinamide metabolism, folate biosynthesis, PPAR signaling pathway and carbon metabolism (Fig. 6a and Supplementary Fig. 8a).

Minor points:

Both the Introduction and Discussion can be expanded.

Response: We appreciate this comment, and have expanded both the Introduction and Discussion as suggested.

As in text:

In the introduction:

ISCs are supported by a niche of accessory cell types to generate mature epithelial cell types, which fuel intestinal renewal, regeneration and development. The niche cells of ISCs including Gli1+, Foxl1+ stromal cells and Paneth cells secrete Wnt molecules as self-renewal factors to support intestinal crypts³⁻⁵.

Notum and Wnt signals were shown to play vital roles in the aging process of intestine marked by reduction of ISC number and alteration of ISC function^{13,14}, but detailed understanding of the signal pathways during the intestinal aging is lacking.

For several tissues with strong energy requirement such as heart and liver tissues, excessive mtDNA mutations result in mitochondrial dysfunction by compromising oxidative phosphorylation and accelerating aging phenotypes²⁰.

In the discussion

During the aging process, mtDNA mutations are observed to be accumulated in aged rodent and human tissues^{21-23,56-59}. We previously showed the accumulation of much more low-frequency (less than 0.5%) mtDNA point mutations in human oocytes during aging, which is linked with impaired blastocyst formation¹⁸. Here, we consistently observed the obvious accumulation of low-frequency point mutations in the aged mouse intestine, implying its important roles in the physiological aging. Notably, the aged intestine of POLG mutator mice shows a similar but faster accumulation of low-frequency mtDNA point mutations than the aged intestine of wild-type mice, providing a suitable model for investigating the roles of mtDNA mutations in the intestinal aging. Using this mouse model, we found that high levels of mtDNA mutations result in NAD⁺ depletion that exacerbates the intestinal aging by ATF5-dependent ISR and its resulting LONP1 upregulation (Fig. 7e).

NAD⁺ depletion is regarded as a common signature of aging^{63,64}, which is observed in many tissues during aging⁶⁵⁻⁶⁸. Identically, we observed NAD⁺ depletion in the intestinal aging caused by mtDNA mutations, supporting its usage as a biomarker for the aging of different tissues. NAD⁺ depletion has been reported to compromise mitophagy^{41,42}, which exacerbates accelerated aging^{41,69-72}. Here, we found the observed mitophagy compromise caused by mtDNA mutations couldn't be restored by NAD⁺ repletion. What is more, the

NAD⁺ depletion activates ISR to regulate the intestinal aging caused by mtDNA mutations, demonstrating its important and conserved roles in the aging process. Thus, our study provides a new mechanism linking NAD⁺ depletion and aging.

Mitochondrial ISR, triggered by mitochondrial stress, modulates protein synthesis and selectively overexpresses a set of stress-responsive genes to activate pathways including one carbon metabolism, serine acid biosynthesis and antioxidant mechanism/redox homeostasis^{45,46,50,51,73-75}. The stress responsive transcription factors, ATF4⁴⁴⁻⁴⁷, ATF5^{48,49} and/or CHOP^{50,51}, are reported to mediate the transcription of mitochondrial ISR genes. We found that high levels of mtDNA mutations trigger an activation of ISR by NAD⁺ depletion. Interestingly, ATF5 but not ATF4 and CHOP is observed to be upregulated in the aged Mut/Mut^{***} intestine, demonstrating a preferentially translation of ATF5 in the intestinal aging caused by mtDNA mutations.

ISR contains components of both UPR_{mt} and endoplasmic reticulum UPR 44, but it doesn't always coincide with UPR_{mt} upon mitochondrial stress. For instance, the inhibition of mtDNA transcription, replication and translation has been reported to trigger ISR but not UPR_{mt}^{47,76}. Here, we found that mitochondrial ISR activation upregulates UPR_{mt} component LONP1 but downregulates UPR_{mt} component HSP60 in the intestinal aging caused by mtDNA mutations, implying a specific regulation of ISR activation on the protein expression of UPR_{mt} components. LONP1 has been described to degrade cytochrome c oxidase subunits IV and V in the complex IV of oxidative phosphorylation⁷⁷⁻⁷⁹. The upregulation of LONP1 provides a possible explanation for the observed mitochondrial dysfunction caused by mtDNA mutations, which further exacerbates the intestinal aging phenotypes.

In this study, we showed that NAD⁺ repletion in vitro and in vivo increases Wnt/ β -catenin signaling as well as the number of LGR5-expressing ISCs in the small intestine. We also found that the inhibition of LONP1 overexpression by siLonp1, ATF5 silence and ISRIB enhance the regenerative potential of aged intestinal crypts. Thus, we found multiple candidates for treating intestinal aging, not just preventing intestinal aging.

REVIEWER COMMENTS

Reviewer #1 (Remarks to the Author):

The authors have addressed most of my concerns in the revised manuscript. They have put a lot of effort to address each individual point and accepted a lot of criticisms. I praise them for the amount of work included in the revised manuscript. Although there is a number of open questions that are left unanswered, the manuscript as it stands now is very interesting, novel and will prompt further research.

Reviewer #2 (Remarks to the Author):

The revision is improved but some previous questions are still not addressed.

Fig 1b. There is little difference in villus length and basal crypt number in WT mice of various ages. It is unclear whether these are indeed aging related changes based on these data, raising the question of the physiological significance of the changes in mitochondrial mutator mice. If villus length and basal crypt number in WT mice do not change with age, these changes in mitochondrial mutator mice are not aging-related changes. The response and new experiments (S1a-d) are not related to villus length and basal crypt number, and therefore do not address this comment.

The mechanism is still shaky. First, the authors claim that NAD depletion leads to UPRmt. How does this work? Among NAD consuming enzymes, SIRT7 has been linked to UPRmt by suppressing the expression of mitochondrial ribosomal proteins (Mohrin, Science 2015; Wang, Cell Metabolism 2023). Is SIRT7 inactivated upon NAD depletion and activated upon NAD repletion? Second, the authors claim that mitochondrial stress upregulates UPRmt component LONP1 by ATF5 but downregulates UPRmt component HSP60 by eIF2 α phosphorylation. This is at odd given that it is well established that HSP60 is induced by ATF5 too (Fiorese, Curr Biol 2016). Third, UPRmt is a protective mechanism, with proteases (such as LONP1) and heat shock proteins (such as HSP60) induced to help cells recover from stress. However, the authors propose that induction of LONP1 is causing the aging phenotypes. Is there any example in the literature support this view?

Reviewer #3 (Remarks to the Author):

The revised manuscript is much improved. There are still a few questions that need to be addressed.

1. The protein loading was not even in Fig. 6b and 6c and supplementary Fig. S8b and the immunostaining results in supplementary Fig. S7 were not good enough, with too much non-specific signals.
2. The aging phenotype of villi remains puzzling. Should Paneth cells form ISCs instead of differentiated epithelial cells?
3. Why is only Wnt4 examined? There are about 20 Wnt molecules. Do mtDNA mutations affect both ISCs and niche cells? Which is more important?

REVIEWER COMMENTS

Reviewer #1 (Remarks to the Author):

The authors have addressed most of my concerns in the revised manuscript. They have put a lot of effort to address each individual point and accepted a lot of criticisms. I praise them for the amount of work included in the revised manuscript. Although there is a number of open questions that are left unanswered, the manuscript as it stands now is very interesting, novel and will prompt further research.

Reviewer #2 (Remarks to the Author):

The revision is improved but some previous questions are still not addressed.

Fig 1b. There is little difference in villus length and basal crypt number in WT mice of various ages. It is unclear whether these are indeed aging related changes based on these data, raising the question of the physiological significance of the changes in mitochondrial mutator mice. If villus length and basal crypt number in WT mice do not change with age, these changes in mitochondrial mutator mice are not aging-related changes. The response and new experiments (S1a-d) are not related to villus length and basal crypt number, and therefore do not address this comment.

Response: We appreciate this comment, and have added experiments (Fig.S1a) to detect the villus length and basal crypt number of small intestine in young (3 months) and old mice (20 months) as suggested. We observed decreased intestinal crypt number and increased villus length in physiological aged mouse intestine, which is consistent with previous reports (PMID: 28297666; 9637773; 30917412; 9751428; 1211303).

As in text:

As well as aged human clinical intestinal samples, small intestine of aged mouse intestine, marked by decreased intestinal crypt number and increased villus length, higher expression of CDKN1A/p21 (a well-known senescence marker) and shorter telomere length (Supplementary Fig.1a-c), accumulates more mtDNA mutations, primarily low-frequency (less than 0.05) point mutations (Supplementary Fig.1d, e).

The mechanism is still shaky. First, the authors claim that NAD depletion leads to UPR^{mt}. How does this work? Among NAD consuming enzymes, SIRT7 has been linked to UPR^{mt} by suppressing the expression of mitochondrial ribosomal proteins (Mohrin, Science 2015; Wang, Cell Metabolism 2023). Is SIRT7 inactivated upon NAD depletion and activated upon NAD repletion? Second, the authors claim that mitochondrial stress upregulates UPR^{mt} component LONP1 by ATF5 but downregulates UPR^{mt} component HSP60 by eIF2 phosphorylation. This is at odd given that it is well established that HSP60 is induced by ATF5 too (Fiorese, Curr Biol

2016). Third, UPR^{mt} is a protective mechanism, with proteases (such as LONP1) and heat shock proteins (such as HSP60) induced to help cells recover from stress. However, the authors propose that induction of LONP1 is causing the aging phenotypes. Is there any example in the literature support this view?

Response: We appreciate this comment, have detected SIRT7 expression in WT/WT*, WT/Mut** and Mut/Mut*** mice at 8 months (New added Fig. S8e), have added data to show mechanism linking NAD⁺ depletion and ISR (New added Fig. S8f), and have clearly rewritten the differences between our findings of a mitochondria-associated aging stress response (MASR) caused by mtDNA mutations and classic UPR^{mt}.

Among NAD⁺ consuming enzymes, SIRT7 downregulation caused by NAD⁺ depletion has been reported to regulate UPR^{mt} by suppressing the expression of mitochondrial ribosomal proteins. We detected the SIRT7 expression in WT/WT*, WT/Mut** and Mut/Mut*** mice at 8 months, and didn't observe the reduction of SIRT7 protein in Mut/Mut*** comparing to WT/WT* mice (New added Fig. S8e), suggesting other mechanisms for regulating ISR. As the elevation of NADH/NAD⁺ ratio caused by electron transport chain inhibition is reported to deplete asparagine and thus activates the ISR via eIF2 α phosphorylation (PMID: 32463360), the observed higher NADH/NAD⁺ ratio (Fig. 2b, c), mitochondrial dysfunction (Fig. S4c, d) and upregulation of eIF2 α -P (Fig. 6b, c) in Mut/Mut*** mice suggested this possibility. We further measured the amino acid abundance in WT/WT*, WT/Mut** and Mut/Mut*** mice, and consistently found the reduction of asparagine in Mut/Mut*** comparing to WT/WT* mice (New added Fig. S8f), supporting that the higher NADH/NAD⁺ ratio caused by mtDNA mutations regulates ISR by asparagine depletion.

We agree that our findings demonstrate a mitochondria-associated aging stress response (MASR) caused by mtDNA mutations, which is different from classic ATF5 dependent UPR^{mt}. UPR^{mt} has been regarded as a self-protective mechanism triggered by mitochondrial import efficiency. It promotes cell survival and mitigates mitochondrial dysfunction to ensure optimal cellular function (29165426; 28687630), and is marked by the increase of LONP1, HSP60 and ClpP. As UPR^{mt} proteases, the upregulation of LONP1 and ClpP helps cells to recover from stress. Different from UPR^{mt}, MASR is a passive stress response triggered by mtDNA mutations during aging process. Upon MASR activation, the result LONP1 upregulation further exacerbates the aging phenotypes of small intestine with accumulated mtDNA mutations, indicating a dual role of LONP1 under stress response. LONP1 has been reported to promote the binding of POLG to mutated mtDNAs, and thus leads to the accumulations of mtDNA mutations. Consistently, we observed the LONP1 upregulation and the accumulations of mtDNA mutations in the aged Mut/Mut*** intestine.

As in text:

We next asked how NAD⁺ depletion regulates ISR? Among NAD⁺ consuming enzymes, SIRT7 downregulation caused by NAD⁺ depletion during aging has been reported to regulate UPR^{mt} by suppressing the expression of mitochondrial ribosomal proteins. We detected the SIRT7 expression in WT/WT*, WT/Mut** and Mut/Mut*** mice at 8 months, and didn't observe the reduction of SIRT7 protein in Mut/Mut*** comparing to WT/WT* mice (Supplementary Fig. 8e), suggesting other mechanisms for regulating ISR. As the elevation of NADH/NAD⁺ ratio caused by electron transport chain inhibition is reported to deplete asparagine and thus activates ISR via eIF2 α phosphorylation⁵⁸, the observed higher NADH/NAD⁺ ratio (Fig. 2b, c), mitochondrial dysfunction (Supplementary Fig. 4c, d) and upregulation of eIF2 α -P (Fig. 6b, c) in Mut/Mut*** mice suggested this possibility. We further measured the amino acid abundance in WT/WT*, WT/Mut** and Mut/Mut*** mice, and consistently found the reduction of asparagine in Mut/Mut*** comparing to WT/WT* mice (Supplementary Fig. 8f), supporting that the higher NADH/NAD⁺ ratio caused by mtDNA mutations regulates ISR by asparagine depletion.

Together, we named this pathway induced by mtDNA mutations, characterized by ATF5 dependent ISR activation and its resulting differential expression of UPR^{mt} components, as mitochondria-associated aging stress response (MASR).

Using this mouse model, we found high levels of mtDNA mutations result in NAD⁺ depletion during the intestinal aging, and subsequently activates MASR, marked by ATF5 dependent ISR activation and its resulting LONP1 upregulation and HSP60 downregulation, to regulate mtDNA mutation-induced aging phenotypes and LGR5 positive ISC exhaustion by impairing Wnt/ β -catenin pathway (Fig. 7e).

UPR^{mt} has been regarded as a self-protective mechanism triggered by mitochondrial import efficiency. It promotes cell survival and mitigates mitochondrial dysfunction to ensure optimal cellular function, and is marked by the increase of LONP1, HSP60 and ClpP. As UPR^{mt} proteases, the upregulation of LONP1 and ClpP helps cells to recover from stress. Different from UPR^{mt}, MASR is a passive stress response triggered by mtDNA mutations during aging process. Upon MASR activation, the result LONP1 upregulation further exacerbates the aging phenotypes of small intestine with accumulated mtDNA mutations, indicating a dual role of LONP1 under stress response. LONP1 has been reported to promote the binding of POLG to mutated mtDNAs, and thus leads to the accumulations of mtDNA mutations. Consistently, we

observed the LONP1 upregulation and the accumulations of mtDNA mutations in the aged Mut/Mut^{***} intestine.

Reviewer #3 (Remarks to the Author):

The revised manuscript is much improved. There are still a few questions that need to be addressed.

1. The protein loading was not even in Fig. 6b and 6c and supplementary Fig. S8b and the immunostaining results in supplementary Fig. S7 were not good enough, with too much non-specific signals.

Response: We appreciate this comment, have added western blotting results with even protein loading (New added Fig. 6b, c and Fig. S8b), and have added better images of anti-Gli1, anti-Notch1 and anti-Foxl1 immunostaining (New added Fig. S7c-d) as suggested.

2. The aging phenotype of villi remains puzzling. Should Paneth cells form ISCs instead of differentiated epithelial cells?

Response: We appreciate this comment, have added experiments (New added Fig.S1a) to detect the villus length and basal crypt number of small intestine in young (3 months) and old mice (20 months), and have rewritten our presentation for Paneth cells as suggested. We observed decreased intestinal crypt number and increased villus length in physiological aged mouse intestine, which is consistent with previous reports (PMID: 28297666; 9637773; 30917412; 9751428; 1211303). Upon aging, intestinal crypt size, villus length, Paneth cell number, and goblet cell number all increase, and the number and regenerative potential of aged ISCs are reduced upon aging in mice (PMID: 28297666; 9637773; 30917412). The mechanism for increased villus length in the small intestine with reduced ISC function (number or regenerative potential), as you mentioned, is a very interesting, important and huge field that has not been solved at present time. We also have rewritten our presentation for Paneth cells as suggested. Moreover, we rewrote the LGR5-expressing ISCs as LGR5 positive intestinal cells, for the possible existence of ISCs without LGR5 or with little LGR5 expression.

As in text:

As well as aged human clinical intestinal samples, small intestine of aged mouse intestine, marked by decreased intestinal crypt number and increased villus length, higher expression of CDKN1A/p21 (a well-known senescence marker) and shorter telomere length (Supplementary Fig.1a-c), accumulates more mtDNA mutations, primarily low-frequency (less than 0.05) point mutations (Supplementary Fig.1d, e).

We noted that LGR5 positive intestinal cells decline sharply at 8 months in Mut/Mut^{***} mice (Fig. 5b), indicating an adverse effect of mtDNA mutations on

the ISCs. NAD⁺ repletion increased the number of LGR5 positive intestinal cells in Mut/Mut^{***} mice (Fig. 5c).

As the activation of Notch signaling has been reported to induce a subset of Paneth cells to gain stem cell features and subsequently proliferate and differentiate into villus epithelial cells, we performed anti-Notch1 IF and did observe the activation of Notch signal in the intestinal crypt base of Mut/Mut^{***} mice at 8 months (Supplementary Fig. 7d)

Upon aging, intestinal crypt size, villus length, Paneth cell number, and goblet cell number all increase, and the number and regenerative potential of ISCs are reduced upon aging in mice. It is interesting that the intestinal villus length increases while ISC number is reduced in the small intestine, requiring further studies.

3. Why is only Wnt4 examined? There are about 20 Wnt molecules. Do mtDNA mutations affect both ISCs and niche cells? Which is more important?

Response: We appreciate this comment, have added experiments to detect additional wnt molecules such as WNT2 and WNT5A secreted from the niche cells in WT/WT^{*}, WT/Mut^{**} and Mut/Mut^{***} intestine (New added Fig. 4d), have added the reason for detecting these wnt molecules, and performed anti-Foxl1 and anti-Notch1 IF in Mut/Mut^{***} mice with NMN or water control to investigate whether mtDNA mutations affect niche cells by NAD⁺ depletion as suggested.

For the requirement of Wnt molecules, such as WNT2, WNT4 and WNT5A, secreted from the niche cells as self-renewal factors to support ISCs, we assayed the expression of these Wnt molecules in the intestinal crypts of WT/WT^{*}, WT/Mut^{**} and Mut/Mut^{***} mice at 8 months. We found all the tested Wnt proteins are decreased in Mut/Mut^{***} mice comparing to WT/WT^{*} (New added Fig. 5d), showing a secretion inhibition of Wnt molecules caused by mtDNA mutations. NAD⁺ repletion rescues the reduction of WNT2 and WNT4 protein in Mut/Mut^{***} mice (Fig. 5d and Fig. S7a, b), implying that mtDNA mutations inhibit the secretion of these two Wnt molecules by NAD⁺ depletion.

We observed Foxl1 downregulation and Notch1 upregulation in Mut/Mut^{***} mice, which could be rescued by NAD⁺ repletion (New added Fig. S7c-e). These results showed mtDNA mutations could also affect niche cells by NAD⁺ depletion. The reduction of ISC and Foxl1 cells in Mut/Mut^{***} mice could largely be rescued by NAD⁺ repletion, suggesting mtDNA mutations could both regulates ISCs and Niche cells by NAD⁺ depletion.

As in text:

For the requirement of Wnt molecules, such as WNT2, WNT4 and WNT5A, secreted from the niche cells including Gli1⁺, Foxl1⁺ stromal, Paneth and myofibroblast cells as self-renewal factors to support ISCs, we assayed the expression of these Wnt molecules in the intestinal crypts of WT/WT^{*},

WT/Mut^{**} and Mut/Mut^{***} mice at 8 months. We found all the tested Wnt proteins are decreased in Mut/Mut^{***} mice comparing to WT/WT^{*} (Fig. 5d), showing a secretion inhibition of Wnt molecules caused by mtDNA mutations. NAD⁺ repletion partially rescues the reduction of WNT2 and WNT4 protein in Mut/Mut^{***} mice (Fig. 5d and Supplementary Fig. 7a, b), implying that mtDNA mutations could inhibit the secretion of these two Wnt molecules by NAD⁺ depletion.

NAD⁺ repletion rescues the Foxl1 downregulation and Notch1 upregulation in Mut/Mut^{***} mice (Supplementary Fig. 7d), suggesting that mtDNA mutations could regulate the function or number of niche cells by NAD⁺ depletion.

REVIEWER COMMENTS

Reviewer #2 (Remarks to the Author):

In the revised manuscript, the authors claim that mtDNA mutation leads to aging defects due to a novel mitochondrial stress response that they term mitochondria-associated aging stress response (MASR), independent of known mitochondrial stress response UPRmt. The authors ruled out the UPRmt because they did not observe the reduction of SIRT7 protein in Mut compared to WT. However, SIRT7 protein levels are not supposed to change because of reduced NAD. Reduced NAD should decrease SIRT7 activity, which is exemplified by p-eIF2a and mitochondrial dysfunction (Ohkubo, Cell Reports 2022; Wang, Cell Metabolism 2023), the changes also observed in Mut mice, suggesting SIRT7 could be mediating these changes in Mut mice. Clearly, to determine the role of SIRT7, a compelling experiment would be taking a genetic approach using SIRT7 KO mice or overexpressing SIRT7. The authors observed reduced asparagine and claimed that this is the cause of NAD depletion mediated ISR. Does asparagine supplement rescue the defect? The authors stated "UPRmt has been regarded as a self-protective mechanism triggered by mitochondrial import efficiency". This is not correct. In fact, UPRmt has been shown to be activated by various mitochondrial stresses, including unbalance of mitochondrial and nuclear proteomes (Houtkooper, Nature 2013), a likely result of mtDNA mutations.

Another remaining issue is LONP1. Simply because LONP1 is upregulated during stress, it is not sufficient to argue LONP1 drives the aging phenotype. It could simply be a reflection of stress. In the literature, LONP1 is associated with improved physiology, in contrast to the authors' claim that LONP1 drives aging. Can the authors find any literature support the idea that LONP1 drives any physiological decline? The authors state "LONP1 has been reported to promote the binding of POLG to mutated mtDNAs, thus leads to the accumulations of mtDNA mutations". This is actually opposite to what was reported by Yang, Nat Cell Biol 2022, showing LONP1 inhibits the binding of POLG to mutated mtDNAs.

Reviewer #3 (Remarks to the Author):

The authors have addressed my concerns in the revised manuscript. T

Reviewer #2 (Remarks to the Author):

In the revised manuscript, the authors claim that mtDNA mutation leads to aging defects due to a novel mitochondrial stress response that they term mitochondria-associated aging stress response (MASR), independent of known mitochondrial stress response UPRmt. The authors ruled out the UPRmt because they did not observe the reduction of SIRT7 protein in Mut compared to WT. However, SIRT7 protein levels are not supposed to change because of reduced NAD. Reduced NAD should decrease SIRT7 activity, which is exemplified by p-eIF2a and mitochondrial dysfunction (Ohkubo, Cell Reports 2022; Wang, Cell Metabolism 2023), the changes also observed in Mut mice, suggesting SIRT7 could be mediating these changes in Mut mice. Clearly, to determine the role of SIRT7, a compelling experiment would be taking a genetic approach using SIRT7 KO mice or overexpressing SIRT7. The authors observed reduced asparagine and claimed that this is the cause of NAD depletion mediated ISR. Does asparagine supplement rescue the defect? The authors stated “UPRmt has been regarded as a self-protective mechanism triggered by mitochondrial import efficiency”. This is not correct. In fact, UPRmt has been shown to be activated by various mitochondrial stresses, including unbalance of mitochondrial and nuclear proteomes (Houtkooper, Nature 2013), a likely result of mtDNA mutations.

Response: We appreciate this comment, have detected the protein expression of eIF2 α -P in Mut/Mut^{***} intestinal crypt cells overexpressing SIRT7 and negative control (New added Fig. S8f) and have revised the un-correct statement “UPRmt has been regarded as a self-protective mechanism triggered by mitochondrial import efficiency” as suggested. We found SIRT7 overexpression inhibits eIF2 α phosphorylation of Mut/Mut^{***} intestinal crypt cells (Supplementary Fig. 8f). Consistent with previous reports (Ohkubo, Cell Reports 2022; Wang, Cell Metabolism 2023), these observations support that NAD⁺ depletion caused by increased mtDNA mutation burden decreases SIRT7 activity to regulate ISR.

In addition, we have detected the protein expression of eIF2 α -P in Mut/Mut^{***} intestinal crypt cells treated with Asn or water control (as below figure a-b). We showed that Asn repletion inhibits eIF2 α phosphorylation of Mut/Mut^{***} intestinal crypt cells, indicating that mtDNA mutations regulate ISR by Asn depletion. We also agree with the reviewer that the SIRT7 is the mainstream mechanism that links NAD⁺ depletion and ISR, and have deleted this inadequate conclusion in this manuscript.

a**b**
a Asn abundance in the small intestine of WT/WT*, WT/Mut** and Mut/Mut*** mice at 8 months ($n \geq 4$ mice per group; $*p < 0.05$, one-way ANOVA test). b Protein expression of eIF2 α -P by western blot analysis in Mut/Mut*** intestinal crypt cells treated with Asn or water control. Relative band densities quantified using ImageJ are shown at right ($n = 3$ mice per group; $**p < 0.01$, paired two-tailed Student's t test).

As in text:

The UPR_{mt} is reportedly activated by various mitochondrial stresses including mitonuclear protein imbalance and mitochondrial import efficiency 61,83, which is marked by increased protein expression of LONP1, HSP60 and ClpP.

Despite no reduction of SIRT7 protein in Mut/Mut*** versus WT/WT* mice (Supplementary Fig. 8g), SIRT7 overexpression inhibits eIF2 α phosphorylation of Mut/Mut*** intestinal crypt cells (Supplementary Fig. 8f). Consistent with previous reports^{59,60}, these observations support that NAD⁺ depletion caused by increased mtDNA mutation burden decreases SIRT7 activity to regulate ISR.

Another remaining issue is LONP1. Simply because LONP1 is upregulated during stress, it is not sufficient to argue LONP1 drives the aging phenotype. It could simply be a reflection of stress. In the literature, LONP1 is associated with improved physiology, in contrast to the authors' claim that LONP1 drives aging. Can the authors find any literature support the idea that LONP1 drives any physiological decline? The authors state "LONP1 has been reported to promote the binding of POLG to mutated mtDNAs, thus leads to the accumulations of mtDNA mutations". This is actually opposite to what was reported by Yang, Nat Cell Biol 2022, showing LONP1 inhibits the binding of POLG to mutated mtDNAs.

Response: We appreciate this comment, have deleted all the presentations "LONP1 drives the aging phenotype", have agreed that LONP1 upregulation is a reflection of stress, have deleted the un-correct statement "LONP1 has been reported to promote the binding of POLG to mutated mtDNAs, thus leads to the accumulations of mtDNA mutations", and

have revised LONP1 upregulation as only one marker of MASR caused by mtDNA mutations as suggested.

As in text:

Mechanistically, increased mtDNA mutation burden exacerbated the aging phenotype of the small intestine through ATF5 dependent integrated stress response activation.

We found that increased mtDNA mutation burden triggers a mitochondria-associated aging stress response by NAD⁺ depletion, characterized by activation of transcription factor 5 (ATF5) dependent integrated stress response (ISR) and resulting Lon protease (LONP1) upregulation.

The UPR_{mt} is reportedly activated by various mitochondrial stresses including mitonuclear protein imbalance and mitochondrial import efficiency 60,83, which is marked by increased protein expression of LONP1, HSP60 and ClpP. We found that increased mtDNA mutation burden triggers a stress response, MASR, during intestinal aging. The activation of MASR, characterized by ATF5 dependent ISR activation and resulting LONP1 upregulation, regulates intestinal aging phenotypes.

The specific upregulation of LONP1 protein, a possible reflection of MASR, provides a candidate marker for intestinal aging caused by increased mtDNA mutation burden.

REVIEWERS' COMMENTS

Reviewer #2 (Remarks to the Author):

The new experiments add clarification to the mechanism. One suggestion on clarification on writing. The authors propose a new term MASR. Since the mechanism is NAD-dependent SIRT7 activation of mitochondrial UPR, it is the best for everyone in the field including the authors to use the same nomenclature and move the field forward. Mitochondrial UPR has been used extensively in the field and do not see why we need to change it and confuse the field.

Reviewer #2 (Remarks to the Author):

The new experiments add clarification to the mechanism. One suggestion on clarification on writing. The authors propose a new term MASR. Since the mechanism is NAD-dependent SIRT7 activation of mitochondrial UPR, it is the best for everyone in the field including the authors to use the same nomenclature and move the field forward. Mitochondrial UPR has been used extensively in the field and do not see why we need to change it and confuse the field.

Response: We appreciate this comment, have changed the title as “NAD+ dependent UPR^{mt} activation underlies intestinal aging caused by mitochondrial DNA mutations”, and have deleted all MASR and use mitochondrial UPR in our manuscript as suggested.